# PfCERLI1 is a conserved rhoptry associated protein essential for *Plasmodium falciparum* merozoite invasion of erythrocytes

Benjamin Liffner [1,8], Sonja Frölich [1,8], Gary K. Heinemann[2], Boyin Liu[3], Stuart A. Ralph [3], Matthew W.A. Dixon [3], Tim-Wolf Gilberger[4,5,6] & Danny W. Wilson [1,7✉]

The disease-causing blood-stage of the *Plasmodium falciparum* lifecycle begins with invasion of human erythrocytes by merozoites. Many vaccine candidates with key roles in binding to the erythrocyte surface and entry are secreted from the large bulb-like rhoptry organelles at the apical tip of the merozoite. Here we identify an essential role for the conserved protein *P. falciparum* Cytosolically Exposed Rhoptry Leaflet Interacting protein 1 (PfCERLI1) in rhoptry function. We show that PfCERLI1 localises to the cytosolic face of the rhoptry bulb membrane and knockdown of PfCERLI1 inhibits merozoite invasion. While schizogony and merozoite organelle biogenesis appear normal, biochemical techniques and semi-quantitative super-resolution microscopy show that PfCERLI1 knockdown prevents secretion of key rhoptry antigens that coordinate merozoite invasion. PfCERLI1 is a rhoptry associated protein identified to have a direct role in function of this essential merozoite invasion organelle, which has broader implications for understanding apicomplexan invasion biology.

[1] Research Centre for Infectious Diseases, School of Biological Sciences, University of Adelaide, Adelaide, SA 5005, Australia. [2] Experimental Therapeutics Laboratory, School of Pharmacy and Medical Sciences, University of South Australia Cancer Research Institute, Adelaide, SA 5005, Australia. [3] Department of Biochemistry and Molecular Biology, Bio21 Molecular Science and Biotechnology Institute, The University of Melbourne, Melbourne, VIC 3010, Australia. [4] Bernhard Nocht Institute for Tropical Medicine, 20359 Hamburg, Germany. [5] Centre for Structural Systems Biology, 22607 Hamburg, Germany. [6] Biology Department, University of Hamburg, 20146 Hamburg, Germany. [7] Burnet Institute, 85 Commercial Road, Melbourne, VIC 3004, Australia. [8] These authors contributed equally: Benjamin Liffner, Sonja Frölich. ✉email: Danny.wilson@adelaide.edu.au

Malaria, caused by infection with *Plasmodium* spp. parasites, results in >400,000 deaths annually with *Plasmodium falciparum* being responsible for the majority of malaria mortality[1]. *Plasmodium* parasites have a complex lifecycle with human infection beginning with transmission of the liver-cell invading sporozoite from a female Anopheline mosquito to the human host. Thousands of daughter merozoites develop in the liver-stages and are released into the blood stream where they invade erythrocytes. In the case of *P. falciparum*, the blood-stage parasite multiplies over the next 48 h, forming 16–32 merozoites that rupture out of the schizont stage parasite and infect new erythrocytes[2]. This asexual blood-stage of the parasite lifecycle causes all disease symptoms.

Erythrocyte invasion is an essential and highly co-ordinated process that begins with reversible attachment of the merozoite to the erythrocyte surface[3]. The merozoite then reorientates such that the apical tip, containing specialised invasion organelles known as the rhoptries and micronemes, contacts the erythrocyte membrane. Invasion ligands are released from the micronemes and rhoptries prior to formation of an irreversible tight-junction that binds the merozoite to the erythrocyte surface. The merozoite then engages an acto-myosin motor complex, anchored in an inner membrane complex (IMC) that lies under the merozoite plasma membrane, to pull the erythrocyte membrane around itself; thus completing invasion[2,4].

The merozoite invasion organelles harbour essential antigens that are the targets of therapeutic interventions aimed at preventing blood-stage parasite replication[3–6]. The contents of micronemes are released first and are responsible for coordinating initial attachment events[7]. The rhoptries, which are the largest of the invasion organelles, adopt a dual club-shape with bulbs further towards the basal side of the merozoite and necks at the apical tip[8]. During invasion, rhoptry neck contents are released first and initiate early interactions with erythrocyte receptors[7,9]. Rhoptry bulb contents are secreted later in the invasion process and are typically involved in parasitophorous vacuole establishment[10].

It is unclear what differentiates the rhoptry bulb from the neck, as no membrane separates them. Because the neck and bulb remain differentiated after plasma membrane fusion, it has been hypothesised that the structure of the rhoptry may be supported by a scaffold, but the components of such a scaffold are yet to be elucidated[11]. Secretion of rhoptry contents may be partially coordinated through changes in the organelle's structure during invasion. The membranes at the tip of the rhoptry neck fuse with the parasite membrane early in invasion and the two rhoptries themselves also fuse beginning at the neck[11]. At the point of erythrocyte entry, the rhoptries fully fuse into a single rhoptry (neck and bulb). The mechanisms leading to rhoptry fusion and secretion are not understood.

Approximately 30 proteins have been linked to the rhoptry[8]. Despite the rhoptries being membrane-bound organelles with dynamic functions during secretion of invasion ligands, there remains only a single protein reported to localise to the luminal face (Rhoptry associated membrane antigen[12]) and two to the cytosolic face (Armadillo repeats only[13], ARO Interacting Protein[14,15]) of the *P. falciparum* rhoptry membrane. Therefore, the drivers of rhoptry secretion and compartmentalisation are unknown. Here we describe a highly conserved *P. falciparum* protein, Pf3D7_0210600 (henceforth named *Plasmodium falciparum* Cytosolically Exposed Rhoptry Leaflet Interacting protein (PfCERLI1)), which lies on the cytosolic face of the rhoptry bulb and is essential for rhoptry secretion and merozoite invasion.

## Results

### PfCERLI1 is conserved and required for blood-stage growth.
PfCERLI1 is a protein of 446 amino acids that is predicted to contain a signal peptide (SP) but no transmembrane domains or glycophosphatidylinositol (GPI) anchor (Fig. 1a). The RNA expression profile and predicted SP of PfCERLI1 has led to speculation that it has a role in merozoite invasion that could be targeted by vaccines[16–18]. PfCERLI1 is highly conserved among *Plasmodium* spp., with >90% identity between *P. falciparum*, *P. reichenowi* and *P. gaboni* orthologues, >75% amino acid identity across human infecting species (including the zoonotic *P. knowlesi*) and >70% when compared to distantly related species (Fig. 1b). To determine whether this conservation was due to a function essential for blood-stage survival, we attempted to disrupt the *Pfcerli1* gene using the selection linked integration-targeted gene disruption system (SLI-TGD) (Supplementary Fig. 1a)[19]. Our attempts to disrupt *Pfcerli1* were unsuccessful, suggesting PfCERLI1 is essential for *P. falciparum* blood-stage growth.

### Loss of PfCERLI1 is inhibitory to blood-stage growth.
In order to analyse protein expression levels and function of PfCERLI1 during blood-stage development, we made transgenic *P. falciparum* parasites with a C-terminal haemagglutinin tagged (HA) PfCERLI1 with control of protein expression achieved through a glucosamine (GLCN) inducible *GlmS* ribozyme knockdown system (PfCERLI1^HAGlmS)[20] (Fig. 1c). The resulting PfCERLI1-^HAGlmS parasites were cloned and analysed by PCR to confirm plasmid integration (Fig. 1d); cloned parasites were used in all subsequent experiments. Using lysates prepared from synchronised blood-stage parasite cultures (harvested at rings, early trophozoites, late trophozoites and schizonts) probed with anti-HA antibodies, we determined that HA-tagged PfCERLI1 was most highly expressed in late schizont stages (Fig. 1e), concordant with previously published transcriptomic data[21].

To validate that the integrated *GlmS* ribozyme could control PfCERLI1 protein expression, we treated PfCERLI1^HAGlmS parasites with 2.5 mM GLCN and quantified changes in HA-tagged protein levels using Western blot. Treatment of synchronous parasites with 2.5 mM GLCN for ~44 h from early ring stage led to a >80% reduction in PfCERLI1 expression, whereas expression of the loading control EXP2 was unaffected (Fig. 1f; Supplementary Fig. 1b,c). In addition, immunofluorescence microscopy analysis revealed a significant reduction in HA labelling across nearly the whole PfCERLI1^HAGlmS parasite population, with a reduction of HA staining by ~97% (GLCN treated vs the mean of untreated parasites, Supplementary Fig. 8c). To determine whether loss of PfCERLI1 protein expression affected parasite growth, early ring stage PfCERLI1^HAGlmS parasites were treated with increasing concentrations (0.125 to 5 mM) of GLCN for 48 h and parasite growth assessed the following cycle. GLCN treatment led to dose-dependent growth inhibition, with 5 mM GLCN leading to an approximately 55% reduction in parasite growth (Fig. 1g). At 2.5 mM GLCN, which we found to have minimal non-specific growth inhibitory activity against 3D7 WT parasites even after 96 h of treatment (Supplementary Fig. 2a), PfCERLI1^HAGlmS blood-stage parasite growth was reduced by ~40% after 48 h treatment. PfCERLI1's high level of conservation and being refractory to disruption suggested an essential role in parasite growth with the stage of protein expression indicating it could be involved in erythrocyte invasion.

### PfCERLI1 has an important role in merozoite invasion.
In order to assess whether PfCERLI1 has a role in merozoite invasion, we co-transfected the PfCERLI1^HAGlmS parasites with a cytosolic green fluorescent protein (GFP) reporter plasmid (PfCERLI1^HAGlmS/GFP) that allowed accurate quantitation of new merozoite invasion events by flow cytometry[22,23]. Since

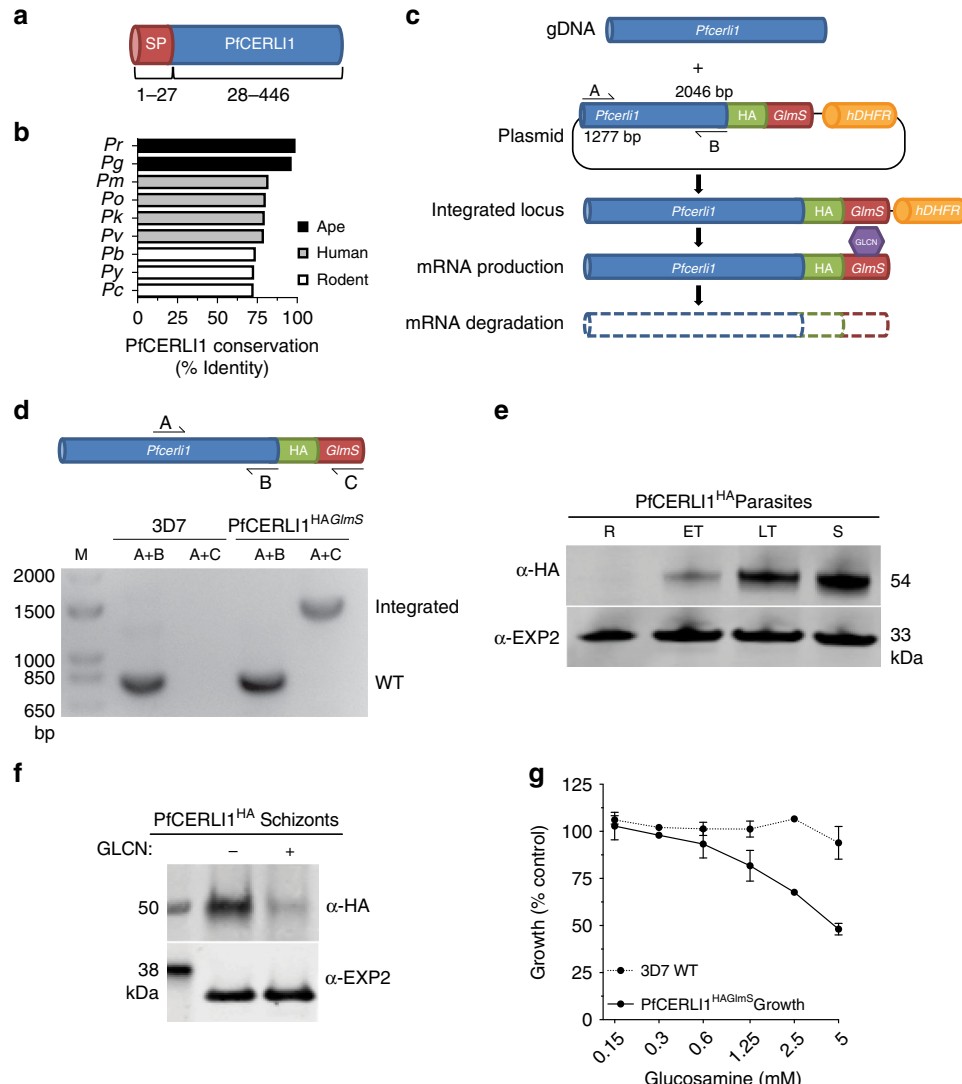

**Fig. 1 Phylogeny of PfCERLI1 (Pf3D7_0210600) and development of genetic tools to investigate function. a** PfCERLI1 is 446 amino acids in length and predicted to contain a signal peptide. **b** The amino acid sequence of PfCERLI1 was compared against *Plasmodium* spp. orthologues by multiple pairwise alignments. *Pr = P. reichenowi, Pg = P. gaboni, Pm = P. malariae, Po = P. ovale, Pk = P. knowlesi, Pv = P. vivax, Pb = P. berghei, Py = P. yoelii, Pc = P. chabaudi.* **c** Schematic representation of the HA-*GlmS* riboswitch system used to study PfCERLI1. A plasmid vector, containing a 3′ flank of the *Pfcerli1* sequence (1277bp-2046bp) with a haemagglutinin (HA) tag and under the control of a *GlmS* ribozyme was transfected into wildtype parasites by 3′ single crossover recombination. Glucosamine (GLCN) binds to the *GlmS* ribozyme, promoting *Pfcerli1* mRNA degradation. **d** Plasmid integration was confirmed by PCR using primers that amplify only WT *Pfcerli1* locus (primer A and B) or primers that amplify only integrated *Pfcerli1*HAGlmS locus (primer A and C). These PCR reactions showed that the majority of parasites had integrated *Pfcerli1*HAGlmS into the correct flanking region (M = size ladder). **e** Western blot of ring (R), early trophozoite (ET), late trophozoite (LT) or schizont (S) stage PfCERLI1HAGlmS lysates probed with anti-HA (PfCERLI1) or anti-EXP2 (loading control) antibodies. **f** Western blot of schizont stage PfCERLI1HAGlmS lysates either in the presence (+) or absence (−) of 2.5 mM GLCN, which was then probed with anti-HA (PfCERLI1) and anti-EXP2 (loading control) antibodies, showing effective knockdown of PfCERLI1 in the presence of GLCN. **g** Synchronous PfCERLI1HAGlmS or 3D7 trophozoite-stage parasites were treated with increasing concentrations of GLCN for 48 h, with the number of trophozoites the following cycle measured to determine knockdown-mediated growth inhibition (*n* = 3 biological replicates. Parasite growth expressed as a % of non-inhibitory media controls, error bars = standard error of the mean (SEM). Source data are provided in source data file). X-axis presented as a log 2 scale for viewing purposes.

PfCERLI1 was expressed only at schizont stages, we limited the amount of time PfCERLI1HAGlms/GFP parasites were exposed to increasing concentrations of GLCN (0.125 to 5 mM) by only treating from early trophozoite stages (24 hpi) for 24 h before assessing parasite rupture and invasion. There was a dose-dependent inhibition of merozoite invasion (Fig. 2a), which was of similar magnitude to the growth inhibition recorded with longer term treatment, strongly suggesting that knockdown of PfCERLI1 restricts parasite growth by preventing invasion.

To determine whether PfCERLI1 knockdown was inhibiting invasion by interfering with merozoite production, the number of merozoites that fully developed per E64 treated PfCERLI1HAGlmS/GFP schizont was counted using light-microscopy of Giemsa-stained thin smears. There was no difference in the number of merozoites or stage of merozoite development evident between GLCN treated and untreated PfCERLI1HAGlmS/GFP parasites (Untreated 18.9 merozoites per schizont; treated 18.8 merozoites per schizont, *p* = 0.5; Fig. 2b). Furthermore, flow cytometry

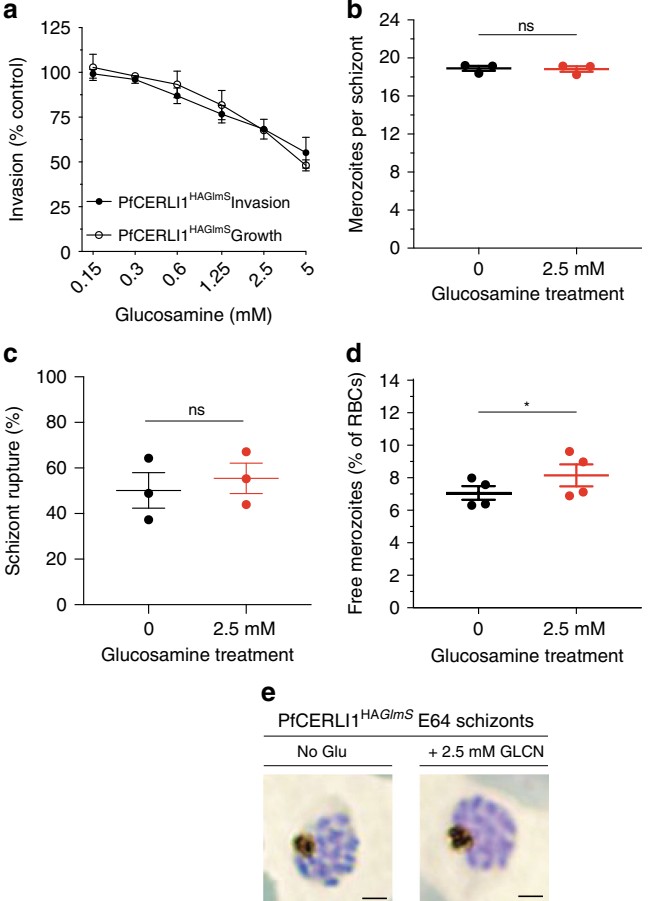

**Fig. 2 PfCERLI1 knockdown does not inhibit merozoite development but does prevent merozoite invasion. a** Flow cytometric detection of GFP-expressing PfCERLI1[HAGlmS/GFP] ring stage parasites after merozoite invasion indicated a direct inhibition of merozoite invasion with protein knockdown (results presented as a % of media control, $n = 4$ biological replicates). PfCERLI1[HAGlmS] Growth is replicated from Fig. 1g for direct comparison between growth and invasion inhibition. *X*-axis presented as log 2 scale for viewing purposes (Source data are provided as a source data file). **b** Mean number of fully segmented merozoites per schizont was determined by microscopy analysis of Giemsa-stained thin blood smears. Smears were blinded and counted with each data point representing the mean number of merozoites per schizont pooled from 20 schizonts, $n = 3$ biological replicates. **c** The percentage of schizont rupture that occurred over a 6-h window either with, or without, GLCN treatment ($n = 3$ biological replicates), **d** The number of free merozoites after GLCN treatment was assessed using flow cytometry, with results presented as % of total erythrocytes ($n = 4$ biological replicates). **e** Giemsa-stained PfCERLI1[HAGlmS] schizonts, either GLCN treated or untreated, matured normally in the presence of E64. Scale bar = 2 μm. All error bars = SEM. *$p$ < 0.05, ns = $p$ > 0.05 by unpaired *t*-test.

analysis also indicated that GLCN treatment did not alter schizont rupture ($p = 0.6$) (Fig. 2c).

When the number of free merozoites between GLCN and non-treated cultures was compared, there were significantly more free merozoites in the GLCN treated cultures ($p < 0.05$) (Fig. 2d). When the number of lost invasion events (lost ring stage parasitaemia) was subtracted from the number of free merozoites, there was no difference between GLCN treated and untreated parasites ($p > 0.99$, Supplementary Fig 2c), indicating that the increase of the free merozoite population was proportional to the loss in successful merozoite invasion events. In addition, GLCN

treated schizonts appeared morphologically normal when Giemsa-stained (Fig. 2e), suggesting PfCERLI1 knockdown does not result in schizont developmental defects. These data indicate that knockdown of PfCERLI1 is associated with a build-up in the number of free merozoites in culture, a pattern consistent with a possible early invasion defect prior to formation of the tight-junction and engagement of the acto-myosin motor.

We also assessed whether knockdown of PfCERLI1 protein expression could lead to reduced or inhibited growth in merozoites that managed to invade a new erythrocyte. We found that there was no difference between the number of early ring stages progressing through to late trophozoite stages (36 h post-invasion) between GLCN treated and untreated cultures, indicating that PfCERLI1 knockdown merozoites that do invade do not have reduced viability (Supplementary Fig. 2b). The similarity between growth and invasion inhibition, the normal development of merozoites and the build-up of free merozoites in the culture media strongly supports a direct role for PfCERLI1 in erythrocyte invasion.

**PfCERLI1 localises to the rhoptry bulb.** Confocal microscopy of immuno-labelled schizonts was used to assess the spatial positioning of HA-tagged PfCERLI1 relative to the micronemal marker Cysteine-rich protective antigen (CyRPA), inner membrane complex marker glideosome-associated protein 45 (GAP45), rhoptry neck marker rhoptry neck protein 4 (RON4) and rhoptry bulb marker rhoptry-associated protein 1 (RAP1) (Fig. 3a). Using thresholded Pearson's correlation coefficient to calculate relative spatial proximity between compartments, semi-quantitative colocalisation analysis indicated that the signal of PfCERLI1 was associated most frequently with the rhoptry bulb marker RAP1 (Fig. 3b). Given the resolution limits of conventional confocal microscopy (~200 nm XY and ~500 nm Z)[24] relative to the size of the double-bulbous rhoptries[11,25], this approach was deemed unsuitable for the analysis of PfCERLI1 subcellular compartmentalisation within the rhoptries. To overcome this, we utilised Airyscan super-resolution microscopy, which revealed both PfCERLI1 and RAP1 displaying a double doughnut-shaped bulbous structure (Fig. 3c). This fluorescence pattern is analogous to the canonical structure of the rhoptry bulb and, combined with the close colocalisation between PfCERLI1 and RAP1 suggested that PfCERLI1 localises on or within the rhoptry bulb.

In order to assess whether PfCERLI1 is on the inside or the outside of the rhoptry bulb, we biochemically characterised the location of this protein in relation to the membrane of the rhoptries using a proteinase K protection assay. This assay determines the subcellular location of a protein as determined by its sensitivity to proteinase K after selective permeabilisation of non-organellar membranes with detergents. Since the rhoptry membrane is resistant to the detergent digitonin, proteins that lie inside the rhoptry are protected from proteinase K cleavage. Digitonin lysis of parasite membranes resulted in proteinase K cleavage of PfCERLI1[HA] as detected by Western blot (Fig. 4a), indicating that PfCERLI1 lies outside the rhoptry and is exposed to the parasite cytosol. By contrast, RAP1, was not degraded significantly by proteinase K treatment, confirming that rhoptry luminal proteins are refractory to digestion. Despite the level of colocalisation between PfCERLI1 and RAP1, this difference in proteinase K sensitivity suggested that PfCERLI1 localised to the cytosolic face of the rhoptry bulb.

Having identified that PfCERLI1 lies on the cytosolic face of the rhoptry bulb membrane, we assessed how HA-tagged PfCERLI1 was bound to the rhoptry membrane through sequential, differential solubilisation of lysed parasite membranes.

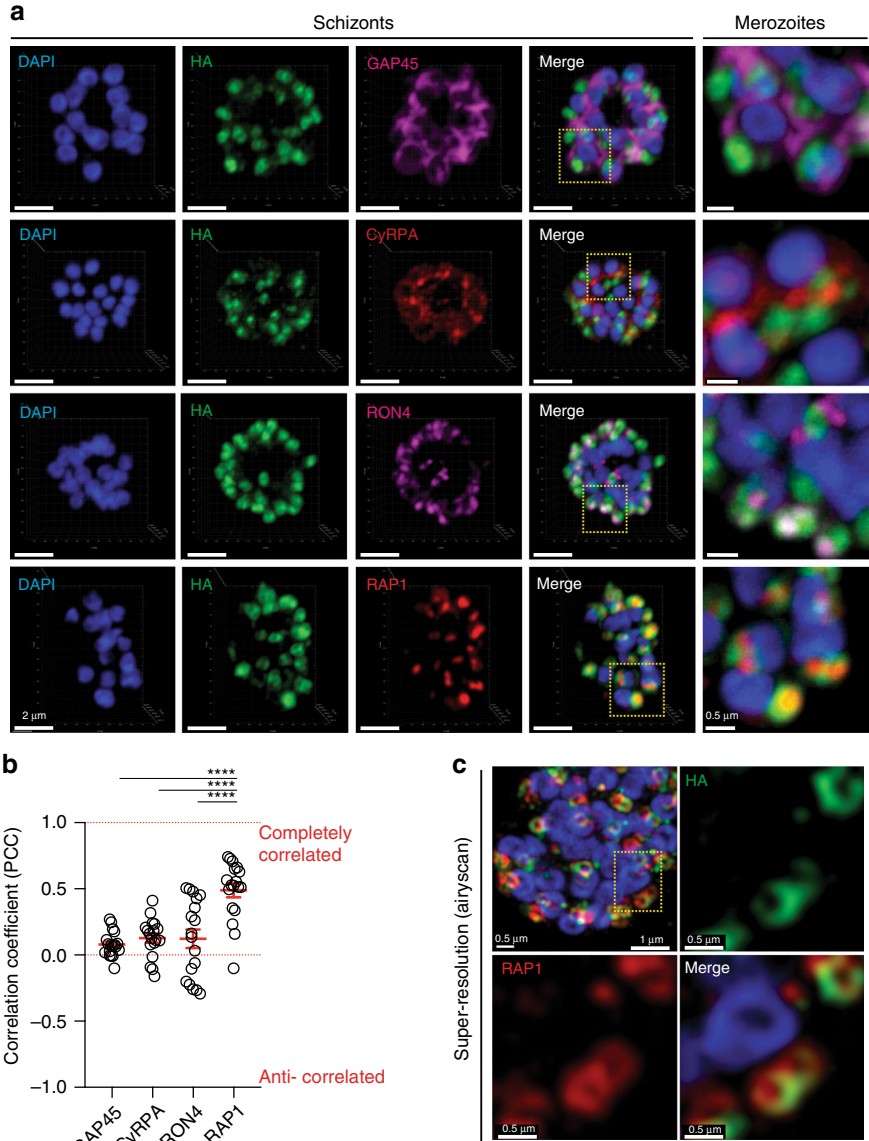

**Fig. 3 PfCERLI1 localises to the rhoptry bulb of merozoites. a** Immunofluorescence microscopy showing 3D reconstructed projections of confocal images with anti-HA (PfCERLI1) in green, colocalising more strongly with the rhoptry bulb marker RAP1 than with the micronemal marker CyRPA, rhoptry neck marker RON4 or the inner-membrane complex marker GAP45. **b** Quantification of the colocalisation, by Pearson's correlation coefficient when PfCERLI1 (anti-HA) staining is defined as the region of interest, with the following merozoite organelle markers: GAP45, CyRPA, RON4, and RAP1. (****$p < 0.0001$ by Analysis of variance (ANOVA), $n = 3$ biological replicates, with 6 merozoite containing schizont images taken from each, error bars = SEM). **c** Maximum intensity projections of super-resolution immunofluorescence microscopy (Airyscan) of PfCERLI1[HAGlmS] schizonts stained with anti-HA and anti-RAP1 antibodies. Yellow box (top left panel) indicates the zoom area for the free merozoite depicted in the other three panels.

Saponin pellets from PfCERLI1[HAGlmS] schizont-stage parasites were hypotonically lysed (to release non-membrane associated proteins), treated with sodium carbonate (Na$_2$CO$_3$; to release peripheral membrane proteins) and Triton X-100 (Tx100; to release integral membrane proteins). The supernatants containing solubilised proteins with the different treatments, as well as the final Tx100 insoluble fraction (containing covalently lipid-linked proteins[26,27]) were used in Western blots (Fig. 4b). PfCERLI1 was detected primarily in the carbonate treatment, indicating that PfCERLI1 is likely to be peripherally associated with the cytosolic face of the rhoptry bulb membrane.

**PfCERLI1 is closely juxtaposed to the rhoptry bulb marker RAP1.** Since biochemical analysis showed that PfCERLI1 associated with the cytosolic face of the rhoptry membrane whereas

RAP1 appears confined to the rhoptry lumen, we used super-resolution microscopy to quantify the degree of physical proximity between PfCERLI1 and RAP1. To do this, we measured the radial positioning of all voxels that define the compartments of interest and plotted the signal intensities for both PfCERLI1 and RAP1 against the length of a vector drawn through the image (Fig. 4c). Using a line scan through a two-dimensional image of fluorescent objects, the distance between PfCERLI1 intensity peaks were significantly wider than that of RAP1 (Fig. 4d, e); further supporting the notion that RAP1 and PfCERLI1 are spatially segregated.

In order to assess whether PfCERLI1 is also associated with markers of the rhoptry neck that lie closer to the apical tip of the merozoite, we triple antibody-labelled schizonts that had been treated with compound 1 (C1), a Protein kinase G inhibitor that

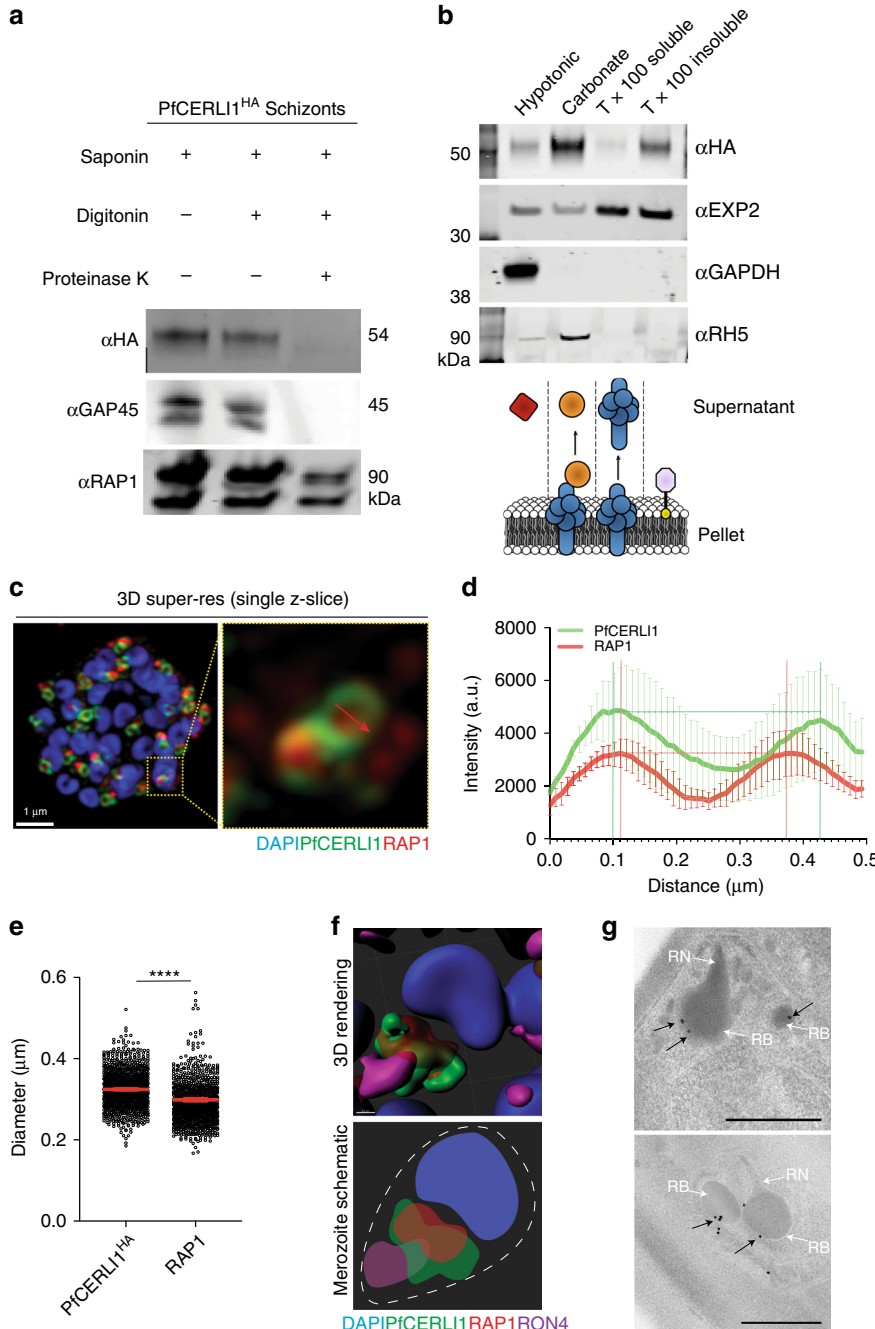

**Fig. 4 PfCERLI1 is peripherally-associated with the cytosolic face of the rhoptry membrane. a** PfCERLI1HA schizont cultures were used for a proteinase K protection assay. Parasites were treated with either saponin alone, saponin and digitonin or saponin, digitonin and proteinase K and probed with anti-HA antibodies (PfCERLI1). GAP45 (inner-membrane complex, exposed to the cytosol) and RAP1 (rhoptry lumen) serving as positive and negative controls, respectively, for proteinase K digestion. Selected images representative of three independent experiments. **b** To determine membrane association, PfCERLI1HA schizont cultures were subjected to a solubility assay. Saponin lysed parasite cultures were hypotonically lysed before being treated with sodium carbonate and then Triton-X-100 (Tx100), with supernatants and the Tx100 insoluble fraction being reserved after each treatment. Resulting samples were probed with anti-HA antibodies (PfCERLI1). The presence of a strong band in the carbonate treatment indicates release of the majority of PfCERLI1 protein into the supernatant with this treatment. Membranes were also stained with anti-EXP2 (transmembrane domain containing), anti-GAPDH (cytosolic) and anti-RH5 (peripheral) solubility controls. Selected images representative of three independent experiments. **c** A single z-slice of the double-bulbous rhoptries displaying the scheme that was used to measure the fluorescence intensity peaks (**d**) and diameter (**e**) of anti-HA and anti-RAP1 staining across the rhoptry (****$p < 0.0001$ by unpaired two-tailed $t$-test, $n = 5$ biological replicates, 1139 rhoptries measured for PfCERLI1HA and 1040 for RAP1, Error bars = SEM). **f** 3D rendered image of a free merozoite showing the nucleus (DAPI) localising at the basal surface, PfCERLI1 (HA) wrapping around RAP1 at the rhoptry bulb and RON4 localising in the rhoptry neck at the far apical tip. **g** Representative image of Compound 1 treated PfCERLI1HAGlmS schizonts that were fixed, labelled with anti-HA antibodies and probed with 18 nm colloidal gold secondary antibodies, before being imaged using transmission electron microscopy. White arrows mark rhoptry bulb (RB) and neck (RN), while black arrows mark PfCERLI1 foci. Scale bar = 500 nm.

prevents parasitophorous vacuole membrane rupture[28], with anti-HA (PfCERLI1), anti-RAP1 (rhoptry bulb), and anti-RON4 (rhoptry neck) antibodies prior to super-resolution microscopy. Enhanced spatial resolution of fully-formed merozoites revealed that RON4 is located closest to the apical tip of the merozoite (Fig. 4f), as expected. Neither PfCERLI1 nor RAP1 exhibited extensive overlap with RON4, with the RAP1 signal enclosed entirely within that of PfCERLI1. These data support that PfCERLI1 lies on the outside of the rhoptry membrane closely juxtaposed to the rhoptry bulb marker RAP1 in the rhoptry lumen, confirming that PfCERLI1 is most closely associated with the outer surface of the rhoptry bulb in mature merozoites. To confirm these findings, we performed transmission electron microscopy (TEM) of immunogold labelled PfCERLI1$^{HAGlmS}$ C1 treated schizonts. Supporting immuno-fluorescence localisation experiments, PfCERLI1 foci localised towards the periphery of the rhoptry bulb (Fig. 4g).

**PfCERLI1 contains lipid binding domains**. To investigate how PfCERLI1 may be binding to the surface of the rhoptry bulb and provide insights on protein function, we used protein structural prediction software Phyre2[29], I-TASSER[30–32], and COACH[33,34]. These analyses suggested that PfCERLI1 contains a C2 domain, which are known lipid binding regions (amino acids ~50-170), a pleckstrin homology (PH) domain that typically binds calcium (amino acids ~250-360; Supplementary Fig. 3a) and an N-terminal alpha helix that is currently annotated as a SP. Outside of these domains PfCERLI1 was predicted to be partially disordered. Like most canonical C2 domains[35], the PfCERLI1 C2 domain was predicted to bind ionic calcium and phospholipids (Supplementary Fig. 3b). Of the models tested against, the PfCERLI1 C2 domain was most structurally similar to the C2 domain of human itchy homolog E3 ubiquitin protein ligase (ITCH) but the ligand(s) of this protein have not been elucidated[36]. The protein modelled to the C2 domain of PfCERLI1 with highest confidence was that of C2-domain ABA-related protein 1 (CAR1) from *Arabidopsis thaliana*, which has been shown to bind phospholipids in a calcium ion dependent manner[37,38]. The PH domain of PfCERLI1 was most structurally similar to the PH domain of Protein kinase B/Akt, and like Akt, was predicted to bind to inositol 1,3,4,5-tetrakisphosphate (IP4) (Supplementary Fig. 3c). IP4 forms the head-group of the phospholipid phosphatidylinositol (3,4,5)-trisphosphate (PIP$_3$), which commonly coordinates protein membrane localisation[39,40]. In addition, PfCERLI1 has previously been predicted to be palmitoylated[41]. Collectively, this suggests that PfCERLI1 contains potentially three different lipid-interacting moieties that may work cooperatively or independently to coordinate the binding of PfCERLI1 to the rhoptry bulb membrane.

PfCERLI1 residing on the cytosolic face of the rhoptry bulb membrane appeared to contrast with its predicted SP, as golgi-mediated trafficking of SP-containing proteins should place them on the luminal side of the golgi-derived rhoptries[25,42]. The putative SP displayed on PlasmoDB[43] is predicted using SignalP-3.0[44], however, the latest version of this program (SignalP-5.0[45]) no longer predicts that PfCERLI1 contains an SP (Supplementary Fig. 4a). To determine whether the putative SP is essential for rhoptry targeting by directing the protein into the secretory system, we generated a PfCERLI1 SP deletion construct (*Pfcerli1$^{NoSP}$*) that lacked the N-terminal 26 amino acids. *Pfcerli1$^{NoSP}$* was then cloned into the pArl1a vector, conjugating PfCERLI1$^{NoSP}$ to GFP and placing *Pfcerli1$^{NoSP}$* under the merozoite-specific *ama1* promoter (Supplementary Fig. 4b). The overexpressed PfCERLI1$^{NoSP}$-GFP fusion protein, like its full-length counterpart[16], localised to the apical tip of merozoites

(Supplementary Fig. 4c); suggesting that the putative SP plays no role in PfCERLI1 trafficking.

**PfCERLI1 knockdown alters rhoptry architecture**. Having identified PfCERLI1's confinement to the cytosolic face of the rhoptry bulb, we investigated whether PfCERLI1 may have a role in invasion ligand processing and distribution, or development of rhoptry architecture. Since a number of rhoptry antigens have been shown to be proteolytically cleaved inside the rhoptry lumen, we reasoned that if loss of PfCERLI1 function inhibited transport of rhoptry antigens to the lumen during biogenesis of the organelle, then we would be able to detect altered protein cleavage patterns in rhoptry antigens that undergo processing within this organelle. To assess this, GLCN treated and untreated PfCERLI1$^{HAGlmS}$ schizont lysates were prepared for Western blot. PfCERLI1 knockdown did not result in any consistent changes in processing of the rhoptry bulb marker RAP1, the rhoptry neck antigen RON4, or a control protein that is a component of the inner membrane complex (GAP45) that all undergo proteolytic cleavage inside the parasite prior to invasion commencing (Supplementary Fig. 1b,c). In addition, total protein levels of both RAP1 and RON4 remained consistent following PfCERLI1 knockdown suggesting there is no relationship between PfCERLI1 and total expression levels of these rhoptry lumen antigens in the parasite.

In order to investigate whether PfCERLI1 caused changes in rhoptry architecture, we visualised rhoptries using TEM (Supplementary Fig. 5). Analysis of C1 matured schizonts failed to reveal any obvious ultrastructural differences between the rhoptries of either GLCN treated or untreated PfCERLI1$^{HAGlmS}$ parasites. While the qualitative EM assessment indicated that PfCERLI1 knockdown was not causing clear changes in rhoptry architecture, we investigated whether there was altered rhoptry antigen distribution with PfCERLI1 knockdown using semi-quantitative super-resolution microscopy. GLCN treated and untreated PfCERLI1$^{HAGlmS}$ (Supplementary Fig. 6a) schizonts were imaged in 3-dimensions to estimate shape (sphericity, oblate (flattened sphere), prolate (elongated sphere)), staining intensity, volume and area for the rhoptry bulb marker RAP1 (Supplementary Fig. 6b). A highly significant decrease in the intensity (50% reduction) and increase in the size of RAP1 staining with PfCERLI1 knockdown was observed. These changes were associated with a significant elongation of the RAP1 signal, but no corresponding change in the RAP1 diameter, indicating that the change in RAP1 staining intensity and size is largely due to elongation of the protein's distribution in the rhoptry (Fig. 5e). Collectively, these changes in intensity and size suggest that knockdown of PfCERLI1 alters the distribution of RAP1 within the rhoptries, and/or rhoptry shape more grossly.

To further investigate the spatial positioning of rhoptry antigens, we quantified spatial proximity between RAP1 and RON4 markers in super-resolved GLCN treated and untreated PfCERLI1$^{HAGlmS}$ schizonts (Fig. 5a–e). We first assessed whether distribution of RON4 changes with PfCERLI1 knockdown as described for RAP1. There was a trend towards greater RON4 intensity (12% increase), probably as a result of a less elongated signal in the rhoptry neck (Supplementary Fig. 6c,d). We next investigated whether changes in RON4 and RAP1 staining altered their localisation relative to each other. Using Mander's correlation coefficient to calculate association frequency between markers, semi-quantitative colocalisation analysis indicated that the proportion of RAP1 signal that overlapped with RON4 after knockdown of PfCERLI1 was significantly reduced (Fig. 5a–d). These data indicate that PfCERLI1 knockdown causes changes in the distribution of antigens within the rhoptry, most notably for

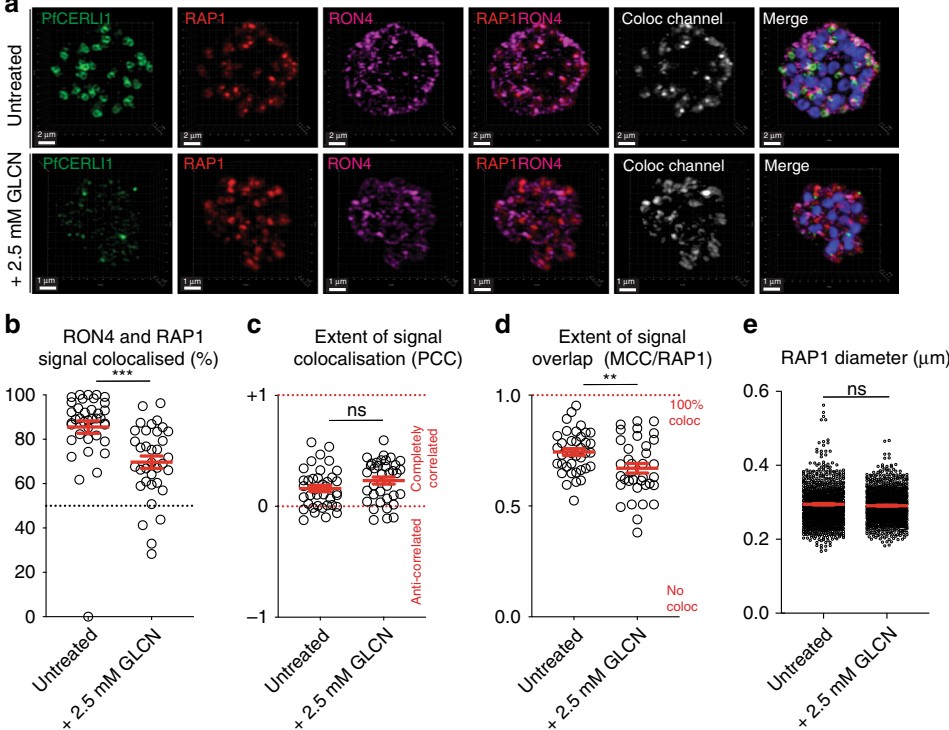

**Fig. 5 PfCERLI1$^{HAGlmS}$ knockdown alters rhoptry antigen distribution. a** 3D reconstructions of super-resolution (Airyscan) micrographs for PfCERLI1$^{HAGlmS}$ schizont cultures that were either left untreated or treated with glucosamine (+2.5 mM GLCN) before being stained with DAPI (nucleus), anti-HA (PfCERLI1), anti-RAP1 (rhoptry bulb) and anti-RON4 (rhoptry neck) antibodies. According to the image analysis pipeline detailed in Supplementary Fig. 7, the (**b**) % of RON4 that colocalised with RAP1, (**c**) Pearson's correlation coefficient (PCC) of the amount of RON4 colocalising with the RAP1 signal, and (**d**) thresholded Mander's correlation coefficient (MCC) of RON4 overlap with RAP1 signal was compared between PfCERLI1$^{HAGlmS}$ knockdown and control parasites. Each data point represents a single schizont image ($n = 5$ biological replicates, 38 schizonts imaged for untreated and 36 for +2.5 mM GLCN schizonts). **e** Rhoptry bulb (RAP1) diameter was measured between untreated and PfCERLI1$^{HAGlmS}$ knockdown (+2.5 mM GLCN) parasites. Each data point represents a single rhoptry ($n = 5$ biological replicates, 1220 rhoptries counted for untreated and 1217 for +2.5 mM GLCN). (ns = $p > 0.05$, **$p < 0.01$, ***$p < 0.001$ by one-way unpaired t-test). All error bars = SEM.

RAP1, and increases the spatial segregation between rhoptry neck (RON4) and bulb (RAP1) markers.

**Loss of PfCERLI1 disrupts merozoite rhoptry antigen function**. After identifying that PfCERLI1 knockdown caused a significant change in the distribution of the rhoptry bulb protein RAP1, we investigated whether knockdown of PfCERLI1 changed apical organelle secretion. Ring-stage cultures were either treated with 2.5 mM GLCN or left untreated before enzymatic removal of erythrocyte surface receptors at trophozoite stages to prevent reinvasion. Schizonts were allowed to rupture, the supernatant collected, and lysates were made from saponin lysed parasite material before analysis by Western blot (Fig. 6a). Blots were probed with anti-RON4 and anti-RH4 antibodies to assess rhoptry secretion, anti-EBA175 to assess microneme secretion, anti-HA to assess PfCERLI1 knockdown and anti-ERC as a loading control. Knockdown of PfCERLI1 did not significantly alter the level of EBA175 (microneme) secretion relative to untreated controls, confirming that release of microneme contents had been triggered for both GLCN and untreated parasites (Fig. 6b). In contrast, quantitation of Western blots revealed a decrease in secreted RON4 and RH4 for PfCERLI1 knockdown parasites relative to untreated controls (Fig. 6b). In addition, RON4 was observed to be more concentrated in the GLCN treated PfCERLI1$^{HAGlmS}$ parasite pellet relative to untreated control parasite material. These data suggest that PfCERLI1 knockdown prevents secretion of essential invasion antigens from

the apical tip of the rhoptry after initiation of the invasion process, providing a probable mechanism by which PfCERLI1 knockdown inhibits invasion.

It has previously been demonstrated that RAP1 is processed by Plasmepsin IX and Subtilisin-like protease 1 (SUB1), beginning as an 84 kDa precursor and being processed to a 67 kDa product that is contained in free merozoites[46–48], with this processing likely required for function. The anti-RAP1 antibody used in this study[49] typically detects three bands, the full-length unprocessed RAP1, a 67 kDa processed RAP1 contained in merozoites, with the middle band suspected to be an intermediate cleavage product or a product of non-physiological procesing[47,48]. While we were unable to detect RAP1 in the supernatant in rhoptry secretion assays (Fig. 6a), we noticed that there was an increase in unprocessed RAP1 for PfCERLI1 knockdown treatments. We quantified the relative abundance of each of the three RAP1 bands present, relative to total RAP1 signal, for both untreated and PfCERLI1 knockdown free merozoites with PfCERLI1 knockdown causing a significant increase in the proportion of unprocessed RAP1 and a corresponding decrease in the proportion of processed RAP1 (Fig. 6c). No change in this trend was observed for 3D7 WT free merozoites in parallel assays treated with 2.5 mM GLCN, suggesting the loss in RAP1 processing is a direct result of PfCERLI1 knockdown (Supplementary Fig. 9a). In addition, RAP1 processing in C1 arrested schizonts was quantified for both PfCERLI1$^{HAGlmS}$ and 3D7 parasites (Supplementary Fig. 9b,c). There was no change in RAP1 processing seen for C1 arrested schizonts with 2.5 mM

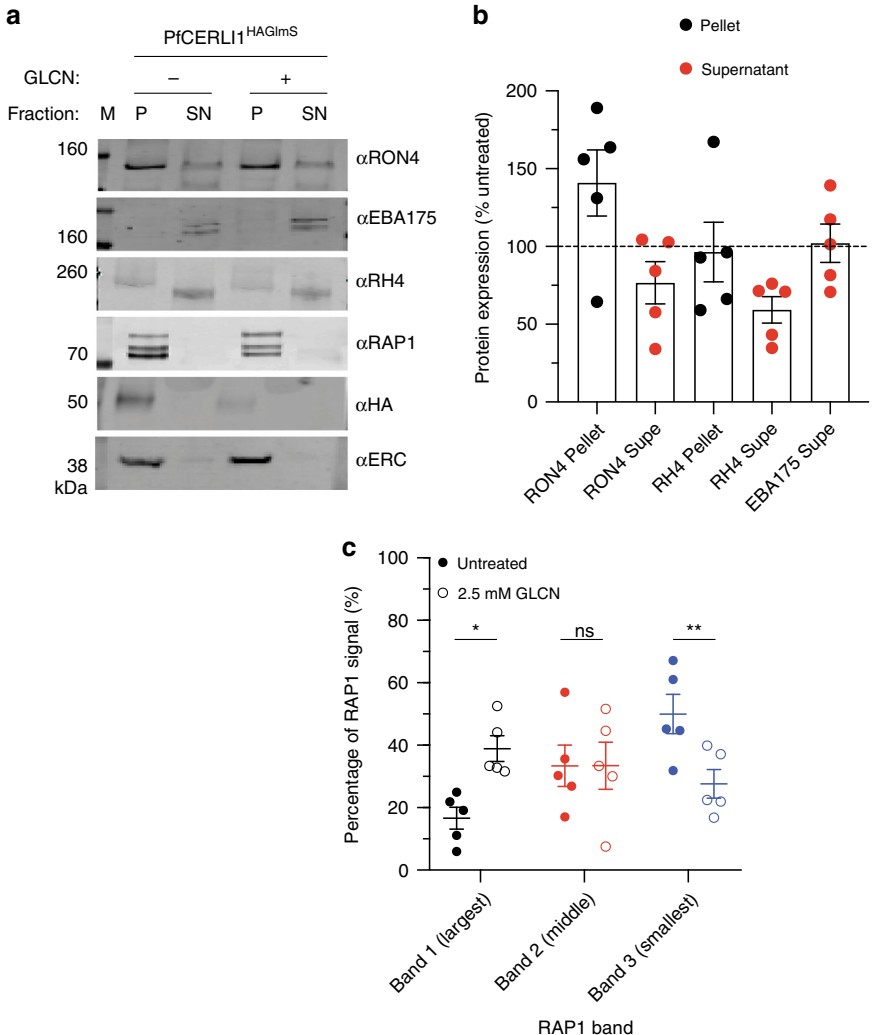

**Fig. 6 PfCERLI1$^{HAGlmS}$ knockdown alters rhoptry antigen processing and secretion. a** PfCERLI1$^{HAGlmS}$ parasites were either treated with 2.5 mM GLCN (+) or left untreated (−), with the erythrocyte surface receptors of the culture cleaved by enzyme treatment to prevent invasion but allow release of merozoite antigens. The parasite lysate (P) and culture supernatant (SN) were stained with anti-RON4 (rhoptry neck), anti-EBA175 (micronemes), anti-RH4 (rhoptry neck), anti-RAP1 (rhoptry bulb), anti-HA (PfCERLI1) and anti-ERC (loading control) antibodies and analysed by western blot. Selected blots representative of five independent experiments (M = size markers). **b** Western blots of rhoptry antigens, and secreted EBA175, were normalised to the loading control (ERC) and quantified, with data represented as +2.5 mM GLCN signal expressed as a % of the signal for the untreated control (n = 5 biological replicates, source data are provided as a source data file). **c** Using the parasite lysates from the rhoptry secretion experiment, each of the three individual RAP1 bands present on the western blots were quantified and presented as a percentage of the total RAP1 signal (n = 5 biological replicates, ns = p > 0.05, *p < 0.05, **p < 0.01 by two-way ANOVA). All error bars = SEM.

GLCN treatment for either PfCERLI1$^{HAGlmS}$ or 3D7 WT parasites, suggesting the inhibition of processing occurs after release of merozoites from the schizont and possibly during the process of invasion. This observation is consistent with prior observations that this cleavage event occurs very late in schizont development or after merozoite egress[50].

## Discussion

Previous studies have identified PfCERLI1 (Pf3D7_0210600) as being a potential vaccine target on the basis of its predicted SP and late schizont stage of expression, a transcription profile typical of proteins involved in merozoite invasion[17,18]. In addition, PfCERLI1 was identified as a member of the 'invadome', an interaction network of putative invasion-associated proteins, and was localised to the merozoite apical tip in this study[16]. Here we show that PfCERLI1 localises to the rhoptries and is confined to the cytosolic face on the rhoptry bulb membrane, a localisation

that would prevent exposure to antibodies during the natural course of invasion. Semi-quantitative super-resolution fluorescence microscopy showed a difference between the mean diameter of PfCERLI1 and the rhoptry bulb marker RAP1 around the bulb structure, indicating they occupy distinct compartments on/within the rhoptry. PfCERLI1 was sensitive to proteinase K after selective membrane solubilisation and RAP1, which appears to lie along the luminal face of the rhoptry bulb and is protected from cleavage, was insensitive, confirming that PfCERLI1 compartmentalises to the cytosolic face of the rhoptry bulb membrane. PfCERLI1 has been shown to be palmitoylated[41], but was predominantly present in the carbonate fraction of the membrane solubility assay (Fig. 4b), suggesting that PfCERLI1 predominantly associates with the cytosolic face of the rhoptry membrane through protein-protein or protein-lipid electrostatic interactions in schizonts. PfCERLI1 was also detected in the Tx100 insoluble fraction. While the contents of this fraction are poorly understood in *Plasmodium*, they have been previously

identified as containing covalently lipid-linked (palmitoylated and myristoylated) members of the IMC[51-53]. C2 domains, as present in PfCERLI1, are predicted to bind calcium and known to form ionic bridges with lipid phosphates in membrane bilayers that are not easily disruptable[54], providing a second potential membrane binding site for PfCERLI1. The presence of two distinct moieties that are predicted to interact with membranes may explain why PfCERLI1 partially solubilises in carbonate buffer while the remainder stays in the Tx100 insoluble fraction.

Our attempts to knock-out *Pfcerli1* proved unsuccessful, confirming a previous random mutagenesis study that identified *Pfcerli1* as essential[55]. However, we successfully integrated a *GlmS*-ribozyme sequence at the 3' end of *Pfcerli1* and were able to knockdown protein expression by >80% leading to >45% inhibition of parasite growth. While we were unable to achieve complete growth inhibition, it is possible that loss of PfCERLI1 function is compensated for by other proteins with similar function or that enough PfCERLI1 remains to enable invasion of some parasites. Indeed, bioinformatic searches identified a second protein, Pf3D7_0405200 (now called PfCERLI2), with similar structure and shared predicted epitopes that has previously been implicated in merozoite invasion[56], providing a possible functional homologue that could compensate for loss of PfCERLI1.

Knockout and knockdown studies implicate PfCERLI1 as having an important role in asexual stage parasite growth and we further confirmed that loss of this protein led to a defect in invasion. Initial investigation of the PfCERLI1 knockdown parasites showed that the number and morphology of merozoites was unaffected by loss of PfCERLI1 and parasites that did invade with PfCERLI1 knockdown developed normally. We determined that PfCERLI1 knockdown led directly to a significant reduction in newly invaded ring-stage parasites and a proportional build up in free merozoites in the culture, possibly through stopping invasion at a point prior to the formation of the tight junction and engagement of the acto-myosin motor. Quantitative assessment of immunofluorescence signal revealed a significant decrease in RAP1 staining intensity with PfCERLI1 knockdown, corresponding to increases in RAP1 volume and area, indicating knockdown of PfCERLI1 changes RAP1 distribution. While changes in area and volume for RAP1 were observed, the diameter of RAP1 rhoptry bulb staining and TEM analysis indicated that there were no large morphological changes in the rhoptry bulb itself. Thus, changes in the ultrastructure of the rhoptry with PfCERLI1 knockdown appear to be minimal and the strongest indication of a modified rhoptry shape achieved to date is increase prolate staining for RAP1, which may indicate a lengthening of the rhoptries that was not noticeable in the single sections of TEM images.

To assess whether the segregation of rhoptry antigens was affected by loss of PfCERLI1, we compared the colocalisation of RAP1 (bulb) and RON4 (neck). Knockdown of PfCERLI1 led to a significant decrease in the proportion of RAP1 overlapping with RON4. Using apical organelle secretion assays, we identified that knockdown of PfCERLI1 resulted in a decrease in secretion of the rhoptry antigens RH4 and RON4 but no change in secretion of the micronemal antigen EBA175. As antigens secreted from the rhoptry neck are essential for merozoite invasion[9,57], disruption of rhoptry neck antigen secretion with PfCERLI1 knockdown points to a role for this protein in successful rhoptry antigen secretion during invasion. Furthermore, when PfCERLI1 was knocked down in free merozoites we saw a clear defect in the processing of the rhoptry bulb antigen RAP1. Given RAP1 processing occurs inside the rhoptry lumen, most likely after merozoite egress, and PfCERLI1 lies on the cytosolic face of the rhoptry bulb, we speculate that aberrant RAP1 processing is likely to be a consequence of general rhoptry dysfunction with PfCERLI1 knockdown.

A recent study by Suarez et al.,[18] reported functional analysis of a homologue of PfCERLI1, which they termed Rhoptry Apical Surface Protein 2 (RASP2), in the related Apicomplexan parasite *Toxoplasma gondii*. They identified that TgRASP2 localises to the cytosolic face of the rhoptry, is essential for tachyzoite invasion of the host cell and confirmed that the C2 domain of this protein binds to the rhoptry membrane through interactions with phospholipids. In the studies of *T. gondii*, TgRASP2 localised along the length of the tachyzoite rhoptries with foci at the rhoptry tip. Parallel studies in *P. falciparum* also identified an essential function for PfCERLI1 (PfRASP2) in invasion using a rapamycin inducible DiCre knockdown system. Although these two studies are highly complementary in the functional analysis of this protein, our quantitative analysis of the localisation of PfCERLI1 differs to that described by Suarez et al., with PfCERLI1 maintaining a clear rhoptry bulb localisation using both fluorescence and immuno-EM localisation in our study and a rhoptry tip localisation reported by Suarez et al. It is possible that PfCERLI1 and TgRASP2 have different localisations/functions, as while these organisms are related, they display markedly different host and cellular tropisms. However, further investigation will be required to tease out the specifics of these differences.

Taking all the data obtained together, we propose that PfCERLI1 has an important role in the function of the rhoptries during invasion, with PfCERLI1 knockdown leading to abnormal distribution, secretion and processing of rhoptry antigens that are likely to be essential for merozoite invasion. While further studies need to be undertaken to determine the fine detail of how PfCERLI1 knockdown causes these changes in rhoptry function, identification of PfCERLI1's direct association with release of rhoptry antigens is a key step in understanding the complex molecular events that control rhoptry secretion during invasion. By understanding how rhoptry secretion is controlled and the key proteins involved, we can identify targets for drug development that will stop merozoites invading and prevent replication of disease-causing parasites. In addition, this study makes extensive use of semi-automated quantitative immunofluorescence microscopy and highlights how this powerful tool can be used to study the process of invasion.

## Methods

**Bioinformatic analyses**. PfCERLI1 (Pf3D7_0210600) and orthologous sequences in *P. reichenow*i (PRCDC_0209500), *P. gaboni* (PGAB01_0208200), *P. malariae* (PmUG01_04021700), *P. ovale* (PocGH01_04019500), *P. knowlesi* (PKNH_0410600), *P. vivax* (PVX_003980), *P. berghei* (PBANKA_0307500), *P. yoelii* (PY17X_0308100) and *P. chabaudi* (PCHAS_0309700) were obtained by searching within the PlasmoDB.org database[43]. Sequence similarities were determined using Geneious 9.1.3 (Biomatters) and performing multiple pairwise alignments using the global alignment with free end gaps alignment algorithm with the Blosum62 cost matrix.

**Continuous culture of asexual stage *P. falciparum***. *P. falciparum* (3D7, 3D7 Δ*Pfcerli1*^HAGlmS and 3D7 Δ*Pfcerli1*^HAGlmS/GFP) parasites were cultured in human O$^+$ erythrocytes (Australian Red Cross blood service)[58]. Parasites were grown in RPMI-HEPES culture medium at pH 7.4 (Gibco) supplemented with 50 μM hypoxanthine, 25 mM NaHCO$_3$, 20 μM gentamicin and either 0.5% Albumax II (Gibco) or 0.25% w/v Albumax II, 5% v/v human serum (Australian Red Cross blood service). Parasite cultures were maintained in an atmosphere of 1% O$_2$, 4% CO$_2$ and 95% N$_2$ at 37 °C.

**Plasmid construction and transfection**. The PfCERLI1^HAGlmS riboswitch transfection vector was prepared from the PTEX150^HAGlmS vector[20,59]. The final 767 base pairs of the 3' end of the *Pfcerli1* genomic sequence (excluding the stop codon) was PCR amplified using the primers *Pfcerli1* 5' F RBW (GGT**AGATCT**CATAT CAAATTTGGTTCTTGAAG) and *Pfcerli1* 3' R RBW (GGT**CTGCAGC**ATCACT ATAGTTGTACATATTTTTGC). The resulting PCR fragment was cloned into the PTEX150-^HAglmS vector using the restriction enzyme cloning sites Bgl II and Pst I (restriction enzyme sites in bold). To generate the cytosolic GFP expressing 3D7

ΔPfcerli1[HAGlmS/GFP] line, the pHGBrHrBl-1/2 GFP plasmid[22] was used, without modification. For PfCERLI1 disruption, a source vector (Pf3D7_1463000 SLI-TGD) was digested with Not I and Mlu I with a 5' Pfcerli1 flank being amplified using the primers (restriction enzyme sites in bold) Pfcerli1 SLI-TGD F (GGT**GCGGCCGC**GATACTCACAACATATTATATCTTGG) and Pfcerli1 SLI-TGD R (GGT**ACGCGT**CATACCTCTATGTGTACTTTGTTCTG)[19]. To construct the PfCERLI1 signal peptide deletion plasmid, Pfcerli1 lacking the signal peptide was amplified using the forward primer Pfcerli1 No SP F (GGT**GGTACC**ATGAGAAACCGTGAGTTATTTC) and the reverse primer Pfcerli1 R (GGT**CCTAGG**ATCACTATAGTTGTACATATTTTGC). The resulting PCR product was restriction digested using KpnI and AvrII and ligated into the pArl1a plasmid, containing the ama1 promoter[60], to generate the Pfcerli1[NoSP] pArl1a plasmid.

P. falciparum 3D7 parasites were transfected using erythrocyte loading[61]. Briefly, uninfected erythrocytes were centrifuged at 1500 rcf for 1 min, before removal of the supernatant and washing in cytomix (0.895% KCl, 0.0017% CaCl₂, 0.076% EGTA, 0.102% MgCl₂, 0.0871% K₂HPO₄, 0.068% KH₂PO₄, 0.708% HEPES). Erythrocytes were then resuspended in cytomix containing 200 µg of ethanol precipitated plasmid DNA, incubated in a 0.2 cm cuvette (Bio-Rad) on ice for 30 min and then electroporated (Bio-Rad) at 0.31 kV with a capacitance of 960 µF. Cells were washed 2× with culture media and transferred to a 10 mL dish containing magnet purified schizont stage parasites (3–5% parasitaemia). Integration of the HA-GlmS plasmid, or presence of the Pfcerli1[NoSP] pArl1a plasmid, was selected for using 3 cycles of 5 nM WR99210 (Jacobus Pharmaceuticals) drug treatment. For generation of the ΔPfcerli1[HAGlmS/GFP] line, PfCERLI1[HAGlmS] parasites were transfected with the an episomal cytosolic GFP expressing plasmid pHGBrHrBl-1/2 and maintenance of this plasmid was selected for using 5 µg/mL blasticidin-S-deaminase HCl (Merck Millipore). Maintenance of the SLI-TGD plasmid was selected for using 5 nM WR99210, with successful integrants then selected for using 400 µg/mL G418 sulphate (geneticin, Thermo Fisher).

**Parasite gDNA extraction and plasmid integration assessment.** To extract parasite gDNA and confirm whether transfected constructs were stably integrated, schizont stage parasite cultures were saponin lysed. gDNA was extracted from saponin lysed parasites using the Wizard® Genomic DNA Purification Kit (Promega) according to the manufacturer's protocol. To confirm integration of Pfcerli1[HAGlmS] into the parasite genome, the Pfcerli1 locus was amplified with the forward primer Pfcerli1 5' F RBW (GGT**AGATCT**CATATCAAATTTGGTTCTTGAAG) and either the Pfcerli1 3' R RBW (GGT**CTGCAG**CATCACTATAGTTGTACATATTTTGC) reverse primer (amplifies wildtype gDNA sequence) or the Glms R (GAAATCCTTACGGCTGTGATCTG) reverse primer (amplifies DNA only upon Pfcerli1[HAGlmS] integration).

**Assessment of in vitro blood-stage growth and invasion.** To determine the number of merozoites produced by each fully formed schizont after knockdown of PfCERLI1 protein expression, thin blood smears were made of D-(+)-Glucosamine hydrochloride (Sigma-Aldrich) (GLCN) treated and untreated schizont stage cultures matured in the presence of E64 (Sigma-Aldrich) to prevent schizont rupture. Smears were methanol fixed and stained with Giemsa (Merck Millipore) before blinded assessment of the number of merozoites by light microscopy (20 individual schizonts). All parasites assessed were fully matured, with individual segmented merozoites visible.

To assess the impact of PfCERLI1 knockdown, 3D7 ΔPfcerli1[HAGlmS] (growth) and 3D7 ΔPfcerli1[HAGlmS/GFP] (invasion) parasites were synchronised to ring stages using sorbitol lysis and assays were set up in 96-well U-bottom plates at 1% parasitaemia and 1% haematocrit in a volume of 45 µL[62]. Five microlitre of 10× concentration GLCN or complete media was added to make a final volume of 50 µL. Assays were stained with 10 µg/mL ethidium bromide (Bio-Rad) in PBS before assessment of parasitaemia using flow-cytometry (BD Biosciences LSR II, 488 nm laser with FITC and PE filters). To assess merozoite development and invasion of 3D7 ΔPfcerli1[HAGlmS/GFP] parasites, GLCN treated and untreated cultures were grown for 36 h, until newly invaded rings were present (0–6 h post-invasion). Flow cytometry data was analysed using FlowJo (Tree Star). The gating strategy used to identify free merozoites and ring-stage parasites is detailed in Supplementary Fig. 10.

**Schizont rupture assay.** To determine whether knockdown of PfCERLI1 altered the ability of schizonts to rupture, PfCERLI1[HAGlmS] parasites were synchronised to ring-stages and set up into duplicate 96-well U-bottom plates, either in the presence or absence of GLCN, as described for growth and invasion assays. Using flow cytometry, the schizont-stage parasitaemia of both treatments was determined at an estimated pre-rupture timepoint for the first plates. After a further 6 h of incubation, the schizont-stage parasitaemia of the second (post-rupture) of the duplicate plates was determined. To determine the percentage of schizont rupture that occurred over the 6-h of incubation the following equation was used:

$$\% \text{ schizont rupture} = \left( \frac{\text{post} - \text{rupture schizontaemia}}{\text{pre} - \text{rupture schizontaemia}} \right) \times 100$$

**Saponin lysis and western blot.** For protein samples, ~10 mL of high parasitaemia culture was lysed with 0.15% w/v saponin for 10 min on ice, parasite material was pelleted by centrifugation before washing once in 0.075% w/v saponin and three times in PBS. In the presence of protease inhibitors (CØmplete, Roche). Parasite lysates were resuspended in reducing sample buffer (0.125 M Tris-HCl pH 7, 4% v/v SDS, 20% v/v glycerol, 10% v/v β-mercaptoethanol (Sigma-Aldrich), 0.002% w/v bromo-phenol blue (Sigma-Aldrich)) and separated by size using SDS-PAGE 4-12% Bis-Tris Gels (Bolt, Invitrogen) at 130 V for 60 min. Proteins were then transferred to a nitrocellulose membrane (iBlot, Invitrogen) at 20 V for 7 min, before blocking the membrane for 1 h at room temperature in Odyssey Blocking Buffer (TBS) (LI-COR Biosciences). Primary (mouse 12CA5 anti-HA (1:4000, Roche), rabbit anti-EXP2 (1:5000,[63]), rabbit anti-GAP45 (1:10000,[64]), mouse anti-RH5 (1:5000), mouse anti-RAP1 7H8 (1:5000,[49]), rabbit anti-RON4 (1:5000), rabbit anti-EBA175 (1:5000[65]), mouse anti-RH4 2E8 (1:2500,[66]), rabbit anti-ERC (1:10000 [67])) and secondary (IRDye ® 800CW goat anti-mouse (1:4000, LI-COR Biosciences), IRDye ® 680RD goat anti-rabbit (1:4000, LI-COR Biosciences)) antibodies were incubated with membranes for 1 h at room temperature (prepared in Odyssey Blocking Buffer (TBS)), before washing in 0.05% v/v PBS Tween followed by a PBS wash after the secondary anti-body incubation. Western blots were visualised using an Odyssey Infrared imaging system (LI-COR Biosciences). Western blot quantification was performed using Image Studio Lite 5.2.5 (LI-COR Biosciences). Uncropped images of all western blots can be found in Supplementary Figs. 11-13.

**Proteinase K protection assay.** For proteinase protection assays[13], three 5 mL aliquots of high-parasitaemia E64 treated schizonts were centrifuged at 440 rcf for 5 min before removal of supernatant and washing in 1× PBS and lysing of uninfected RBCs using saponin. The following treatments were undertaken: One tube with SOTE buffer (0.6 M sorbitol, 20 mM Tris HCl pH 7.5, 2 mM EDTA) alone. A second tube with SOTE plus 0.02% w/v digitonin (Sigma-Aldrich) incubated for 10 min at 4 °C prior to washing in SOTE buffer. A third tube with digitonin treatment followed by digestion with 0.1 µg/µL Proteinase K (Sigma-Aldrich) in SOTE for 30 min at 4 °C. Proteinase K was inactivated by adding 50 µL of 100% v/v tri-chloroacetic acid followed by a PBS wash. All tubes were then resuspended in 500 µL acetone before cells were pelleted, the supernatant removed, and the pellet used for Western blot analysis of proteinase K sensitivity.

**Protein solubility assay.** For protein solubility assays[13], Saponin-lysed pellets from 10 mL of high-parasitaemia E64 treated schizonts were resuspended in 100 µL dH₂O, snap-frozen at −80 °C four times, passed through a 26-gauge needle 5 times to disrupt the parasite membrane and then centrifuged at 15,000 rcf for 10 min with the water-soluble supernatant reserved. The pellet was washed twice in dH2O and once in 1× PBS before resuspension in 0.1 M sodium carbonate (Na₂CO₃) for 30 min at 4 °C with the supernatant reserved. Following carbonate treatment, samples were resuspended in 0.1% v/v Triton-X-100 for 30 min at 4 °C with the supernatant reserved and the resulting pellet washed and resuspended in 1× PBS. Samples were analysed by Western blotting.

**Sample preparation for microscopy.** Untreated and GLCN treated cultures (approximately 10 mL of 3% parasitaemia) of Compound 1 arrested PfCERLI1[HAGlmS] late-stage schizonts were concentrated by centrifugation at 1700 rpm for 3 min, washed in PBS and fixed with 4% v/v paraformaledehyde (PFA, Sigma-Aldrich), 0.0075% v/v glutaraldehyde (pH 7.5, Electron Microscopy Sciences) solution for 30 min at room temperature with gentle shaking. Fixed parasite suspensions were adjusted to 1% haematocrit prior to adhering onto 0.01% poly-L-lysine (Sigma-Aldrich) coated #1.5 H high-precision coverslips (Carl Zeiss, Oberkochen, Germany) for 1 h at room temperature before permeabilistion with 0.1% v/v Triton-X-100 for 10 min. Coverslips were incubated with fresh blocking solution (3% w/v BSA-PBS, 0.05% w/v Tween-20) for at least 1 h. Primary antibodies (Chicken anti-HA (Abcam), mouse anti-RAP1[49], mouse anti-CyRPA 8A7[68], rabbit anti-RON4[69], rabbit anti-GAP45[64]) were diluted 1:500 in antibody diluent (1% w/v BSA-PBS, 0.05% v/v Tween-20) and added to coverslips for 1 h at room temperature or overnight at 4 °C. Cells were washed three times with PBS-Tween-20 (0.1% v/v) and incubated with goat anti-chicken/mouse/rabbit Alexa Fluor coupled secondary antibodies (488 nm, 594 nm, 647 nm) (Life Technologies) diluted 1:500 for 1 h at room temperature. After secondary antibody incubations, coverslips were washed three times with PBS-Tween-20 (0.1% v/v) then post-fixed with 4% w/v PFA for 5 min. PFA was washed off and coverslips dehydrated in ethanol at 70% v/v (3 min), 95% v/v (2 min) and 100% v/v (2 min) before being allowed to air dry and mounted on slides with ProLong® Gold antifade solution (refractive index 1.4) containing 4', 6-diamidino-2phenylindole, dihydrochloride (DAPI) (ThermoFisher Scientific). Once the mountant had cured for 24 h, cells were visualised by either conventional confocal (Olympus FV3000) or Zeiss LSM 800 Airyscan super-resolution microscopy (Carl Zeiss, Obekochen, Germany).

**Confocal microscopy.** Conventional confocal microscopy was performed on Olympus FV3000 fluorescent microscope (Olympus) equipped with a ×100 MPLAPON oil objective (NA 1.4) using the 405 nm, 488 nm, 561 nm, and 633 nm lasers. Z-stacks were acquired with a step size of 0.43 µm using a sequential scan (scan zoom = 10, without line averaging).

**Airyscan super-resolution microscopy**. Sub-diffraction microscopy was performed on a Zeiss LSM800 AxioObserver Z1 microscopy (Carl Zeiss, Obekochen, Germany) fitted with an Airyscan detector and a Plan-Apochromat ×63 (NA 1.4) M27 oil objective. TetraSpeck™ Fluorescent Microspheres (Life Technologies) were mounted on #1.5H coverslips (Carl Zeiss, Oberkochen, Germany) with a density of ~2.3 × 10¹⁰ particles/ml and used as the 200 nm bead calibration sample to correct for chromatic and spherical aberrations. The super-resolution reconstructions of multi-labelled PfCERLI1$^{HAGlmS}$ schizonts were acquired, sequentially in four channels, as follows: channel 1 = 633 nm laser, channel 2 = 561 nm laser, channel 3 = 488 nm laser, channel 4 = 405 nm laser, with exposure times optimised from positive and negative control samples (HA blocking peptide, or omitted primary antibody) to avoid saturation. Three-dimensional (3D) Z-stacks were acquired at a pixel resolution of 0.04 μm in XY and 0.16 μm intervals in Z using piezo drive prior to being Airyscan processed in 3D using batch mode in ZEN Black (Zeiss).

**Diameter measurements of super-resolved rhoptries**. For further spatial exploration of the colocalised signals, 3D volumes captured with super-resolution microscopy after Airyscan processing were imported into ZEN Blue (Zeiss) for object-based colocalisation analysis. In this approach, volumetric colocalisation relies on manual identification of structures of interest and a subsequent measurement of their fluorescence intensity curves. Each rhoptry bulb immunolabelled with anti-RAP1 and anti-HA antibodies was manually quantified by a researcher blinded to experimental conditions, and the organelle diameter measured by drawing a vector through the centre of these structures and plotting the fluorescence intensities for the green and red channel against the length of the vector. Fluorescence intensity profiles of overlapping subcellular structures were then analysed in successive single sections from an image stack representing the two RAP1 and PfCERLI1$^{HAGlmS}$ labelled structures. Fluorescence curves between the two antipodal points on the surface of the ring-like organelle were exported as.csv Excel files and plotted against the distance in nanometres. Only rhoptries with homogeneously stained ring-like structures containing no gaps in RAP1 or PfCERLI1$^{HAGlmS}$ labelling and >100 nm in size were used for diameter measurements. Schizonts densely packed with merozoites with overlapping rhoptries were excluded from scoring. More than 800 individual organelles were scored and data from biological replicates plotted in Prism for comparison of diameters between untreated and glucosamine treated parasites.

**Immunogold labelling and transmission electron microscopy**. Schizont stage parasites were Percoll purified and fixed in 1% glutaraldehyde in RPMI-HEPES buffer. Samples were dehydrated in increasing concentrations of ethanol and embedded in LRGold resin. Ultrathin sections were cut and then labelled with anti-HA antibody (Mouse monoclonal clone 12CA5, Roche, diluted 1:500) and then with 18 nm Colloidal Gold (AffiniPure Goat Anti-Mouse IgG, product 115-215-166, Jackson ImmunoResearch, diluted 1:20). Sections were then post-stained with uranyl acetate and lead citrate and viewed using a Talos L120C Transmission Electron Microscope.

**Rhoptry and microneme secretion assays**. Apical organelle secretion assays were modified from a previous protocol[70]. Briefly, synchronous ring-stage cultures were obtained using sorbitol and either treated with 2.5 mM GLCN or left untreated and incubated in standard culture conditions for 24 h until early trophozoite stage. Cultures were then enzyme treated with neuraminidase (0.067 U/mL), chymotrypsin (1 mg/mL) and trypsin (1 mg/mL). Parasites were then allowed to rupture over the following 24 h. Following rupture, cultures were centrifuged at 13,000 rcf for 10 min in a benchtop centrifuge with some culture supernatant reserved, stored on ice, and the rest removed. The remaining pellet was then also placed on ice and saponin lysed to form a lysate containing the parasites that did not rupture and free merozoites, with this pellet and the culture supernatant analysed by Western blot.

**Statistical analysis**. Graphs and statistical analyses were performed using GraphPad PRISM 7 (GraphPad Software Inc.). In all figures where p-values were calculated, the corresponding statistical test is listed in the figure legend along with the number of experiments results are pooled from.

**Reporting summary**. Further information on research design is available in the Nature Research Reporting Summary linked to this article.

## Data availability
The data that support the findings of this study are available from the corresponding author upon reasonable request. The source data underlying Figs. 1g, 2a, 6b and Supplementary Fig. 2a are provided as a source data file. Unprocessed Gel and Blot images are provided in Supplementary Fig. 11 for blots in Figs. 1d–f, 4a, b, Supplementary Fig. 12 for blots in Fig. 6a, and Supplementary Fig. 13 for blots in Supplementary Fig. 1b, 9a-c.

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

## Acknowledgements

We thank the Australian Red Cross Blood Bank for the provision of human blood. We thank Prof. Alan Cowman for provision of CyRPA, RON4, EBA175 and GAP45 antibodies, Dr. Paul Gilson for EXP2 antibodies, A/Prof. Wai-Hong Tham for RH4 antibodies, and Prof. Leann Tilley for ERC and GAPDH antibodies. We also thank Dr. Paul Gilson for the PTEX150^HAGlmS transfection vector. Confocal microscopy was performed at Adelaide Microscopy, University of Adelaide, and super-resolution microscopy was performed at the Centre for Cancer Biology Cytometry Facility, The University of South Australia. We especially thank Dr. Jane Sibbons from Adelaide Microscopy for assistance with confocal microscopy. Electron microscopy was performed at the Bio21 Institute Advanced Microscopy Facility, The University of Melbourne (www.microscopy.unimelb. edu.au). For provision of the SLI-TGD vector, we thank Dr. Tobias Spielmann. We thank Dr. Brad Sleebs for Compound 1. We thank Arne Alder and Sarah Lemcke for help with SLI-TGD transfection and parasite culture. This work was supported by funding from the NHMRC (Project Grant APP1143974, D.W.W.), University of Adelaide Beacon Fellowship (D.W.W.), DAAD/Universities Australia joint research co-operation scheme (T. G., D.W.W., B.L.), Australian Government Research Training Program Scholarship (B. L.), South Australian Commonwealth Scholarship (B.L.).

## Author contributions

Study design and planning: D.W.W., B.L., S.F., M.W.A.D., S.A.R. and T.G. Performed experiments and generated reagents: B.L., S.F., D.W.W., G.K.H., S.A.R. and B. Liu. Data analysis: S.F, B.L. and D.W.W. Manuscript writing: B.L., S.F, T.G. and D.W.W. Manuscript was drafted with input from all authors.

## Competing interests

The authors declare no competing interests.
