## [Peer Review File · Nature Communications]

Reviewers' Comments:

Reviewer #1:

Remarks to the Author:

This is an interesting paper in which the authors describe a function for the *P. falciparum* protein PfCERL1 in rhoptry protein secretion and invasion. PfCERL1 is a protein that was previously implicated as being part of an interaction network of invasion-associated proteins. Here the authors used a genetic approach to study PfCERL1 function, and demonstrate that knockdown of the protein using a ribozyme tag leads to an invasion defect. They further perform extensive cell biological and biochemical experiments to study the localization of PfCERL1 in relation to known invasion proteins in the merozoite and provide some evidence that PfCERL1 knockdown may lead to changes in rhoptry protein secretion. This work is novel because very little is known about the function of rhoptry proteins in *P. falciparum*, and nothing is known about PfCERL1. While the work has the potential to demonstrate an important advance in our understanding of this *P. falciparum* protein, some controls are missing, making it difficult to know how much to trust the conclusions being proposed.

Major suggestions for improvement:

1. I do not see any evidence showing that the PfCERL1-HA-GlmS strain actually has HA and GlmS tags in the PfCERL1 locus. How do we know that the HAGlmS plasmid was integrated correctly and that PfCERL1 is the protein being studied? This should be confirmed by PCR for the integrated locus and/or southern blot, and included in the manuscript as supplementary material. On a related point, is the PfCERL1HAGlmS parasite a clone?
2. Figure 2c: While this panel shows that there is no difference in the late stage parasitemia for no drug versus 2.5 mM glucosamine treatment, 2% is a high parasitemia if it is being measured post-schizont rupture, as implied. To address whether glucosamine harms schizonts it would be more helpful to show data where the schizontemia is measured pre-rupture and post-rupture. If glucosamine has no effect on schizont rupture, then you would expect to see the schizontemia go down equally in both conditions.
3. There is a mismatch between the figures and the legends in the supplementary files. It appears the legend listed to be for the first figure is actually for the second figure. I cannot tell if the legend currently listed under figure 2 is meant for the first figure, because it doesn't entirely make sense for that figure either.
4. The Western blot in the supplementary material (second figure) should not have a title implying that GLCN specifically knocks down PCERL1. Instead all that can be said is that it specifically knocks down an HA-tagged protein. To confirm knockdown of PfCERL1, one would need to have a specific antibody or to do RT-PCR. This brings up the point that there is no confirmation of appropriate integration of the construct in PfCERL1 locus (see first comment).
5. Figure 2d. The conclusion from this experiment that GLCN leads to impaired ring formation prior to formation of the tight junction is not supported by the results that are presented. Four data points are presented, but are these technical replicates or biological replicates? The error bars seem too small to represent biological replicates. How many of these free merozoites are actually invasive? The approach presented seems unnecessarily complex and difficult to interpret. To make the claim that a very early stage of invasion is affected by GLCN, it would be more straightforward to perform a free merozoite invasion assay and measure the ring parasitemia 5 minutes after adding the merozoites. Alternatively, the authors could perform an attachment assay using Cyt-D to measure if merozoite attachment is impaired for parasites grown in GLCN.

6. The penultimate sentence in the results section titled "PfCERL1 has an important role in merozoite invasion" says " We found that there was no difference between the number of early ring stage parasites that progress through to late troph stages between GLCN treated and untreated cultures.... (Supp Fig. 2B). However, supplementary Fig 2B is a graph showing something about free merozoites. This needs to be corrected and clarified both in the figures and the results.

7. The super-resolution microscopy in Fig 3c for would easier to interpret if the images were presented as individual stains followed by a merged image, as in 3a. In the current layout, it is difficult to determine how the authors are concluding that "both PfCERL1 and RAP1 display a double doughnut structure". To really demonstrate that PfCERL1 is localized in the rhoptry bulb, a more appropriate experiment might be immuno-EM using the HA antibody. Was that attempted in the context of the TEM experiments, and if so what did it show? This seems an important point given the difference in proteinase K sensitivity between PfCERL1 and RAP1, and the authors extensive efforts to define the relative localization of these two proteins.

8. In Figure S6 where 3D super-resolution microscopy is used to measure merozoite proteins what does n=5 mean in these experiments? 5 merozoites? 5 schizonts, where all merozoites within a schizont were averaged? The error bars for some of the measurements deemed to be significantly different are exceedingly small, which makes me worry that these are only technical replicated and there may be batch effects. In this same experiment, how do you control for variation in developmental stage of a "schizont"? In the representative images in S6a, based on the number of merozoites that can be distinguished it looks like the GLN treated schizont is older than the untreated one. Perhaps these measurements need to all be done on schizonts with equal numbers of merozoites, and biological replicates should include merozoites from multiple schizonts.

9. In the western blot in Figure 5F-G there is not a convincing difference between the amount of RON4 observed in pellet/supernatant in the untreated versus treated parasites. Also, I believe there is some evidence to suggest that RH5 is actually a micronemal protein (not rhoptry), so this is not an ideal protein to examine in this experiment. Given the very mild phenotype seen for RON4 it would be helpful to include some other rhoptry neck and/or bulb proteins in this experiment in order to provide more support for the conclusion that PfCERL1 knockdown impairs rhoptry content release.

10. If there is truly a decrease in RON4 release from the rhoptry, this would imply that the moving junction would not form normally, as the RON complex needs to be introduced into the uninfected RBC for formation of the moving junction. To more directly address this question I suggest looking directly at formation of the moving junction in invading merozoites that have or have not been treated with GLCN, or at localization of the components of the moving junction during invasion (RON complex and AMA1). This would provide more compelling evidence that problems with rhoptry antigen discharge are the reason for impaired invasion. If this is deemed out of the scope of this work at this time, then I suggest that the conclusion that PfCERL1 plays a key role in controlling the distribution and secretion of rhoptry proteins during invasion should be dampened and presented as more of a working hypothesis in the discussion.

Minor comments:

1. Fig 1c can be deleted and instead the reference for SLI-TDG paper can just be provided, especially since the technique was unsuccessful for PfCERL1 KO in this case.

2. Figure panel 1d would be more informative if it showed a schematic of the integration plasmid and site of homology where recombination is expected to occur, along with binding location of primers used

to confirm correct integration.

3. Figure 1g: Does n=3 refer to technical replicates or biological replicates? If biological replicates, please define. Please indicate the ring parasitemia detected in the media control condition in the legend

4. For Figure 2a, does n=4 indicate technical replicates or biological replicates? If biological replicates, please indicate how these are defined (different parasite clones, same clone run on different days, etc?). Please indicate the % ring parasitemia detected at 24 hours in the control condition in the legend.

Reviewer #2:

Remarks to the Author:

The study "PfCERLI1, a conserved rhoptry associated protein essential for invasion by Plasmodium falciparum merozoites" by Liffner et al, combines a conditional protein knockdown strategy with confocal and super-resolution microscopy, together with extensive biochemical characterization to study the essentiality and function of PfCERLI1. The authors show that PfCERLI1 is essential and is important for erythrocyte invasion. Using super-resolution microscopy and stepwise biochemical fractionation analyses, the authors also show that PfCERLI1 is peripherally associated with the cytosolic face of rhoptries. Using a GLMS-based knockdown approach together with flow cytometric quantification of released merozoites, the authors determine a role for PfCERLI1 during invasion. In an attempt to decipher the mechanism through which PfCERLI1 exerts its invasion phenotype, the authors use semi-automated image analysis and conclude a change in rhoptry structure in PfCERLI1 knock down parasites. Finally, the authors observe a reduction in rhoptry protein secretion in PfCERLI1 knock down parasites, implicating PfCERLI1 in rhoptry secretion.

The novelty of the work lies not only in the fact that a rhoptry associated protein was found on the cytosolic face, with a probable function in rhoptry secretion, but also in the use of semi-automated quantification techniques of super-resolution images. The extensive biochemical characterization of where and how PfCERLI1 is associated with the rhoptries, and the careful use of control experiments, is commendable. While this study is of great interest to the field studying host cell invasion, certain key experiments would strengthen the conclusions that the authors make.

1. The authors see an 80% reduction in protein levels by Western Blot by 48h. This is of course a bulk measure. How does the reduction look in individual cells? An IFA to look at depletion of PfCERLI1 levels in individual cells is important. Do different cells have different levels of protein knock-down? Or, do all cells show the same level of reduction? This result has important implications for the phenotype.

2. The authors conclude from the increased number of free merozoites in culture supernatants that a defect in invasion prior to tight junction formation exists (lines 175-176). The authors also note that microneme secretion (as measured by EBA175 secretion) is not compromised in these parasites. Microneme secretion of certain proteins (including EBA175) establish secondary interactions with the RBC. It would be very useful to decipher exactly at what step, erythrocyte invasion is compromised in the PfCERLI1 knock-down parasites. Several questions arise: Do these merozoites attach to RBCs and then fall off? What degree of attachment do they have?

Measurement of attachment of the PfCERLI1 KD merozoites to erythrocytes, by attachment assays AND/OR video microscopy to score for deformation of the RBCs by these merozoites would be valuable additions.

3. It has been previously observed that treating parasites with cytochalasin-D blocks entry but does

not prevent rhoptry secretion - whorls of RAP1 can be seen in uninvaded RBCs (see Fig 6, Riglar et al (2011) Cell Host & Microbe). If an IFA were performed on the RBCs, which the PfCERLI1 knock down merozoites failed to invade, would one see rhoptry proteins aberrantly secreted into the RBCs? If this experiment gives a negative result compared to the positive control, it would strengthen the idea that PfCERLI1 knock downs do not form a tight junction.

Minor points:

1. Colocalisation analysis indicated that PfCERLI1 is most closely associated with the rhoptry bulb protein RAP1. The authors, through extensive biochemical analysis, then show that PfCERLI1 is peripherally associated with rhoptries. Together with this, immuno-EM studies would prove beyond doubt that PfCERLI1 is at the outer leaflet of rhoptries. The authors should consider doing this experiment.
2. Lines 140-141 "By contrast, GLCN treatment did not lead to significant growth inhibition in control 3D7 parasites." Is this data available to be shown in the supplement? The authors say that 2.5 mM GLCN has "minimal non-specific growth inhibitor activity" - a quantitative statement for this would be beneficial.
3. In Fig. 2d, the number of free merozoites are only marginally higher (1-2 %?) than the controls? Do these fully account for the 40% growth defect seen in the knock down?
4. In supplementary Figures 1 and 2, the legends have been interchanged - please correct this oversight.
5. Line 156 - what is an early trophozoite? This is a long window of development - the number of hours should be mentioned for an accurate descriptor.
6. Compound 1 - The authors make use of Compound-1 to arrest schizonts at a very mature stage without letting them undergo egress. Some descriptive sentences about what Compound-1 is, and some references for its use in prior literature are appropriate and necessary.

Response to Referees Letter for Manuscript: 'PfCERLI1, a conserved rhoptry associated protein essential for invasion by *Plasmodium falciparum* merozoites.' by Liffner B., et al.

Reviewer 1 major comments:

- 1. I do not see any evidence showing that the PfCERLI1-HA-GImS strain actually has HA and GImS tags in the PfCERLI1 locus. How do we know that the HAGImS plasmid was integrated correctly and that PfCERLI1 is the protein being studied? This should be confirmed by PCR for the integrated locus and/or southern blot, and included in the manuscript as supplementary material. On a related point, is the PfCERLI1HAGImS parasite a clone?**

Response: We apologise for this oversight. This is something that we should have included in the original submission and we should have also clearly indicated that the lines were cloned. We have now included an integration check PCR in Figure 1. Using a reverse primer specific to the *GImS* sequence we show that the construct is integrated into the genome of transfected parasites but not 3D7 parental parasites.

Following the generation of the PfCERLI1HAGImS transgenic parasites, the parasites were cloned, and all results presented in this manuscript used clonal transfectant parasite lines. All clones were PCR checked for integration of the HAGImS plasmid prior to use in downstream experiments.

To reflect this addition to Figure 1, the figure legend has been modified as follows:

*(line 1036-1040) "Fig 1. Plasmid integration was confirmed by PCR using primers that would amplify only WT *Pfcerli1* locus (primer A and B) or primers that would amplify only integrated *Pfcerli1*^{HAGImS} locus (primer A and C). These PCR reactions showed that the majority of parasites in culture had integrated the *Pfcerli1*^{HAGImS} into the correct flanking region (M = size ladder)."*

In addition, we have added to the main text the following statement to clarify the use of cloned, PCR-checked lines for all experiments:

(line 122-125) "The resulting PfCERLI1^{HAGImS} parasites were cloned. Clonal lines were analysed by PCR confirming integration (Fig. 1d) of the plasmid that encodes a glucosamine inducible riboswitch and the HA-tag used in all subsequent experiments."

The following section has also been included in the methods to clarify this:

(line 571-578) “**Parasite gDNA extraction and plasmid integration assessment**

To extract parasite gDNA and confirm whether transfected constructs were stably integrated, schizont stage parasite cultures were saponin lysed. gDNA was extracted from saponin lysed parasites using the Wizard® Genomic DNA Purification Kit (Promega) according to the manufacturer’s protocol. To confirm integration of *Pfcerli1*^{HAGlms} into the parasite genome, the *Pfcerli1* locus was amplified with the forward primer *Pfcerli1* 5’ F RBW (GGTAGATCTCATATCAAATTTGGTTCTTGAAG) and either the *Pfcerli1* 3’ R RBW (GGTCTGCAGCATCACTATAGTTGTACATATTTTTGC) reverse primer (amplifies wildtype gDNA sequence) or the *Glms* R (GAAATCCTTACGGCTGTGATCTG) reverse primer (amplifies DNA only upon *Pfcerli1*^{HAGlms} integration).”

2. **Figure 2c: While this panel shows that there is no difference in the late stage parasitemia for no drug versus 2.5 mM glucosamine treatment, 2% is a high parasitemia if it is being measured post-schizont rupture, as implied. To address whether glucosamine harms schizonts it would be more helpful to show data where the schizontemia is measured pre-rupture and post-rupture. If glucosamine has no effect on schizont rupture, then you would expect to see the schizontemia go down equally in both conditions.**

Response: We agree that the data provided previously did not fully cover the course of schizont rupture that would allow us to be sure there was no significant rupture defect. To determine this, we set up synchronous ring-stage cultures in duplicate 96-well plates that were either GLCN treated or untreated. The schizontaemia of the first plate was determined by flow cytometry prior to schizont rupture. The schizontaemia of the second plate was assessed after 6 hours of incubation when the majority of rupture events were likely to have occurred. Measuring schizontaemia prior to rupture and then again six hours later, allowed us to calculate how much rupture had occurred over that 6 hour period. Our results showed that there was no difference between GLCN treated and untreated parasites, suggesting that neither GLCN, nor *PfCERL1* knockdown, significantly inhibited schizont rupture.

We have replaced the previous Figure 2c with the results of this experiment and have updated both the figure legend and methods sections as follows:

(line 1068-1071) “**Fig 2. (c)** The percentage of schizont rupture that occurred over a 6-hour window either with, or without, GLCN treatment. Each data point represents the mean of duplicate wells from three biological replicates, ns = $p > 0.05$ by unpaired t-test, error bars = SEM)”

We have added the following information in the methods to describe how changes in schizontaemia were assessed accurately by flow cytometry:

(line 604-614) **“Schizont rupture assay**

To determine whether knockdown of PfCERLI1 altered the ability of schizonts to rupture, PfCERLI1^{HAGImS} parasites were synchronised to ring-stages and set up into duplicate 96-well U-bottom plates, either in the presence or absence of GLCN, as described for growth and invasion assays. Using flow cytometry, the schizont-stage parasitaemia of both treatments was determined at an estimated pre-rupture timepoint for the first plates. After a further 6 hours of incubation, the schizont-stage parasitaemia of the second (post-rupture) of the duplicate plates was determined. To determine the percentage of schizont rupture that occurred over the six-hours of incubation the following equation was used:

$$\% \text{ schizont rupture} = \left(\frac{\text{post-rupture schizontaemia}}{\text{pre-rupture schizontaemia}} \right) \times 100$$

Additionally, to show that glucosamine treated schizonts don't display any gross morphological defects in regard to merozoite segregation, we have included images of PfCERLI1^{HAGImS} schizonts that were matured in the presence of E64 as Figure 2e. These images show a similar number of fully formed merozoites in both glucosamine treated and untreated parasites. The text has been changed to reflect this inclusion as follows:

(line 174-186) *“When the number of free merozoites between GLCN and non-treated cultures was compared, there was significantly more free merozoites in the GLCN treated cultures ($p < 0.05$) (Fig. 2d). When the number of lost invasion events (lost ring stage parasitaemia) was subtracted from the number of free merozoites, there was no difference between GLCN treated and untreated parasites ($p > 0.99$, Supplementary Fig 2c), indicating that the increase of the free merozoite population was proportional to the loss in successful merozoite invasion events. Additionally, GLCN treated schizonts appeared morphologically normal when Giemsa stained (Fig. 2e), suggesting PfCERLI1 knockdown does not result in schizont developmental defects. These data indicate that knockdown of PfCERLI1 is associated with a build-up in the number of free merozoites in culture, a pattern consistent with a possible early invasion defect prior to formation of the tight-junction and engagement of the acto-myosin motor.”*

- 3. There is a mismatch between the figures and the legends in the supplementary files. It appears the legend listed to be for the first figure is actually for the second figure. I cannot tell if the legend currently listed under figure 2 is meant for the first figure, because it doesn't entirely make sense for that figure either.**

Response: The order of these figures were accidentally mixed but this has now been rectified. Additionally, the order of the panels for Supplementary figure 2 were reversed but this too has been fixed. The Figure legend for Supplementary Figure 2

now reads:

(supplementary line 194-209)“**Supplementary Figure 2. Influence of GLCN on wildtype parasite growth, PfCERL1 knockdown on parasite development post-invasion, and the contribution of invasion inhibition to quantified free merozoites. (a)** 3D7 wildtype parasites were treated with increasing concentrations of GLCN for 96 hours, to assess off-target growth inhibitory effects of GLCN (parasite growth expressed as a of media control (n = 5, error bars = SEM). **(b)** Early PfCERL1^{HAGImS/GFP} Ring-stage parasites were treated with glucosamine (2.5 mM GLCN) or left untreated (media). Immediately after invasion the knockdown treatment was removed (washout) or left on (no washout) and ring-stage parasitaemia was determined by flow cytometry. Schizont-stage parasitaemia was then determined 36 hours later by flow cytometry with results reported as the percentage of successfully formed rings (% rings retained) that had survived to schizont-stages (n=2) (error bars = SEM). **(c)** Quantification of free merozoites as presented in Figure 2d, but the data points in the 2.5 mM GLCN treatment represent the number of free merozoites expected after subtraction of additional merozoites that failed to invade relative to untreated controls. (n=4) (error bars = SEM).”

- 4. The Western blot in the supplementary material (second figure) should not be have a title implying that GLCN specifically knocks down PCERL1. Instead all that can be said is that it specifically knocks down an HA-tagged protein. To confirm knockdown of PfCERL1, one would need to have a specific antibody or to do RT-PCR. This bring up the point that there is no confirmation of appropriate integration of the construct in PfCERL1 locus (see first comment).**

Response: As suggested in response (1) we included the integration check PCR to show that the Pfcerli1^{HAGImS} construct has been specifically integrated into these parasites. We believe that this, in conjunction with the presence of a singular anti-HA western blot band at the predicted protein size that is reduced in the presence of glucosamine, is substantial evidence to say that PfCERL1 is being specifically knocked down. We also note that the use of gene-edited protein tags (i.e. HA, GFP) to track changes in protein expression for specific proteins of unknown function, in the absence of RT-PCR data or specific antibodies, is a common level of evidence used in recent studies of similar topics (Rudlaff et al., 2019, Nat. Comms.; Counihan et al., 2017, eLIFE; Suarez et al. 2019, Nat. Comms). We also demonstrate in Supplementary Figure 1 b & c, that glucosamine treatment does not change expression levels for a panel of merozoite antigens in schizonts, supporting that knockdown is specific for the PCR confirmed tagged PfCERL1 protein.

5. Figure 2d.

-The conclusion from this experiment that GLCN leads to impaired ring

formation prior to formation of the tight junction is not supported by the results that are presented.

As detailed in the following, we have taken the reviewers concerns onboard and attempted to confirm the indication that knock-down of PfCERLI1 prevents invasion prior to tight junction formation experimentally but were unable to do so satisfactorily due to technical limitations of the assay. Here, we have chosen to reword the relevant sections in the results and discussion as follows:

Results

(line 174-18) “When the number of lost invasion events (lost ring stage parasitaemia) was subtracted from the number of free merozoites, there was no difference between GLCN treated and untreated parasites ($p>0.99$, Supplementary Fig 2c), indicating that the increase of the free merozoite population was proportional to the loss in successful merozoite invasion events. Additionally, GLCN treated schizonts appeared morphologically normal when Giemsa stained (Fig. 2e), suggesting PfCERLI1 knockdown does not result in schizont developmental defects. These data indicate that knockdown of PfCERLI1 is associated with a build-up in the number of free merozoites in culture, a pattern consistent with a possible early invasion defect prior to formation of the tight-junction and engagement of the acto-myosin motor.”

Discussion

(line 461-469) “Knockout and knockdown studies implicate PfCERLI1 as having an important role in asexual stage parasite growth and we further confirmed that loss of this protein led to a defect in invasion. Initial investigation of the PfCERLI1 knockdown parasites showed that the number and morphology of merozoites was unaffected by loss of PfCERLI1 and parasites that did invade with PfCERLI1 knockdown developed normally. We determined that PfCERLI1 knockdown led directly to a significant reduction in newly invaded ring stage parasites and a proportional build up in free merozoites in the culture, possibly through stopping invasion at a point prior to the formation of the tight junction and engagement of the acto-myosin motor.”

-Four data points are presented, but are these technical replicates or biological replicates? The error bars seem too small to represent biological replicates.

Response: The data points here are the mean from 4 biological replicates, where each biological replicate value is the mean of technical triplicates. To make the extent of replication clearer, we have altered the figure legend as follows:

(line 1060-1061) “Each data point represents the mean of triplicate wells from four biological replicates”

--How many of these free merozoites are actually invasive?

The reviewer raises the question of how many free merozoites are invasive. The non-GLCN treated parasites provide an indication of the number of non-invasive merozoites in the normal (no PfCERLI1 KD) cultures, since each of these merozoites that failed to invade can be considered non-invasive. Invasive merozoites would be those that have gone on to form rings in this assay for the non-GLCN treated culture. In the presence of GLCN, there was a significant decrease in ring stages (viable parasites) which was proportional to the increase in the number of free merozoites (non-viable) in the culture. Simply, as an increasing number of merozoites fail to invade with PfCERLI1 knock-down (i.e. they become non-viable due to loss of this proteins function) we see a proportional build-up in the number of free merozoites.

-To make the claim that a very early stage of invasion is affected by GLCN, it would be more straightforward to perform a free merozoite invasion assay and measure the ring parasitemia 5 minutes after adding the merozoites.

The reviewer suggests two potential assays to interrogate this further. The first assay suggested is the use of free merozoites (Boyle and Wilson et al. PNAS, 2010) and measuring attached merozoites within 5 minutes of mixing free merozoites with RBCs. Corresponding author Wilson, who established this method and used it across a number of studies, has significant experience with these assays. A major problem with this approach is that it is not possible to standardise the number or non-PfCERLI1 mediated merozoite viability between two different parasite populations (i.e. GLCN and no-GLN treated parasite populations) after filtration through a 1.2 μ M filter. In effect, the stressors of the filtration step along with the exact age and number of schizonts cannot be standardised effectively between treatments of two different parasite populations. The free merozoite method is only suited to looking at different treatments (i.e. antibodies, drugs) of the same population of free merozoites.

-Alternatively, the authors could perform an attachment assay using Cyt-D to measure if merozoite attachment is impaired for parasites grown in GLCN.

A second approach suggested by the reviewer is to look at whether PfCERLI1 knock-down reduces the number of merozoites attached to the RBC membrane of cytochalasin D treated cultures. Cyt-D is an inhibitor of actin polymerisation and has been shown to stop merozoite invasion after formation of the tight-junction, leading to an increase in the number of merozoites stuck to the RBC surface with treatment of purified merozoites + RBCs for 2 minutes prior to fixation of this interaction (Riglar et al., Cell Host & Microbe, 2011). We attempted to develop a similar assay for use in general culture conditions which is significantly more feasible for comparing non-GLCN and GLCN treated parasites (graph below). We treated tightly synchronised (42-46 hr age range) schizonts for 6 hours with 0.5 μ M CytoD (higher concentrations caused reduced parasite rupture, n=4 biological replicates). This concentration of CytoD efficiently inhibited formation of ring stage parasites, lead to a proportional build-up of free merozoites, but was not associated with a significant or proportional increase in the number of bound merozoites relative to non-CytoD treated controls as measured by flow-cytometry (Boyle and Wilson et al. PNAS, 2010). Similarly, knock-down of PfCERLI 1 led to a decrease in ring stages, a proportional increase in the number of free merozoites, but did not significantly increase the number of bound

merozoites relative to control. These data indicate that PfCERL1 knock-down did not cause an obvious reduction in the number of bound merozoites in the GLCN + CytoD treated cultures relative to CytoD treatment alone.

However, since CytoD itself did not cause a major increase in the bound merozoite population relative to untreated control cells, we have not included this data in the main manuscript. Instead, we present the data here below to highlight that we attempted this experiment as suggested by the reviewer. We believe that the longer treatment times required for these culture-based experiments lead to the loss of CytoD treated bound merozoites over time compared to assays that use 2 minute treatment of purified merozoites (Riglar et al. Cell Host & Microbe, 2011). We speculate that this may be because the purified merozoite experiments were treated with CytoD 2 minutes after mixing merozoites with RBCs, allowing CytoD to block the acto-myosin motor function halfway through invasion for many merozoites. In the experiments attempted by us, treatment with CytoD commenced before release of

merozoites and throughout invasion. Therefore, the acto-myosin motor may never fully engage and bound merozoites could only be attached through the tight-junction in all treatments, which may not be strong enough to maintain this interaction over the course of several hours in culture and subsequent washes.

In the interests of moving this review forward, we have chosen to reword our conclusions for this set of data and highlight that the proportional build-up of free merozoites would fit a model where invasion is stopped early in the process, as outlined above.

6. The penultimate sentence in the results section titled “PfCERL1 has an important role in merozoite invasion” says “ We found that there was no difference between the number of early ring stage parasites that progress through to late troph stages between GLCN treated and untreated cultures.... (Supp Fig. 2B). However, supplementary Fig 2B is a graph showing something about free merozoites. This needs to be corrected and clarified both in the figures and the results.

Response: This was rectified with the figure legend changes in relating to comment 3. Supplementary Figure 2b is now the graph that corresponds to this statement, and

that section of the figure legend now reads as follows:

(supplementary line 199-209) **“(b) Early PfCERL1^{HAGImS/GFP} Ring-stage parasites were treated with glucosamine (2.5 mM GLCN) or left untreated (media). Immediately after invasion the knockdown treatment was removed (washout) or left on (no washout) and ring-stage parasitaemia was determined by flow cytometry. Schizont-stage parasitaemia was then determined 36 hours later by flow cytometry with results reported as the percentage of successfully formed rings (% rings retained) that had survived to schizont-stages (n=2) (error bars = SEM).”**

- 7. The super-resolution microscopy in Fig 3c for would easier to interpret if the images were presented as individual stains followed by a merged image, as in 3a. In the current layout, it is difficult to determine how the authors are concluding that “both PfCERL1 and RAP1 display a double doughnut structure”.**

Response: As suggested by the Reviewer, we have modified Figure 3c to the layout suggested. We have also tried to clarify this in the figure legend by adding the following:

(line 1087-1090) **“(c) Maximum intensity projections of super-resolution immunofluorescence microscopy (Airyscan) of PfCERL1^{HAGImS} schizonts stained with anti-HA and anti-RAP1 antibodies. Yellow box (top left panel) indicates the zoom area for the free merozoite depicted in the other three panels.”**

- 8. To really demonstrate that PfCERL1 is localized in the rhoptry bulb, a more appropriate experiment might be immuno-EM using the HA antibody. Was that attempted in the context of the TEM experiments, and if so what did it show? This seems an important point given the difference in proteinase K sensitivity between PfCERL1 and RAP1, and the authors extensive efforts to define the relative localization of these two proteins.**

Response: Based on the data we have generated, we believe that PfCERL1 lies on the outside of the rhoptry membrane, while PfRAP1 remains in the lumen as previously reported. Our evidence for this is that in proteinase K protection assays PfCERL1 is susceptible to cleavage (exposed) while RAP1 is not susceptible to cleavage (contained in the rhoptry membrane, Fig. 4a). In addition, in Figures 4 c-e, we show that PfCERL1 staining has a wider diameter than RAP1, a staining pattern that supports the model that PfCERL1 lies on the outside of the rhoptry membrane while PfRAP1 lies on the inside of the membrane. In addition, we have subsequently performed immuno-EM on these parasites, confirming that PfCERL1 localises to the rhoptry bulb. Additionally, PfCERL1 was concentrated on the outside of the rhoptry bulb membrane. However, TEM immunogold labelling has a couple of inherent limitations that are widely known. First is that the signal will be detected ~30nm away from the protein it is bound to and so we cannot thoroughly conclude whether or not

it is on the cytosolic face of the rhoptry membrane from these images alone. Second is that there is inevitably some level of background staining evident, and so it is not possible to quantitatively assess localisation of immunogold labelling as we have done using fluorescence labelling experiments. Nevertheless, we clearly and reproducibly see immunogold labelling of HA tagged PfCERLI1 juxtaposed to the cytosolic face of the rhoptry bulb membrane. The ImmuoEM data has been included in Figure 4g, with the following modifications to the legend of Figure 4:

(line 1115-1119) **“Fig 4. (g) Representative image of Compound 1 treated PfCERLI1^{HAGImS} schizonts that were fixed, labelled with anti-HA antibodies and probed with 18 nm colloidal gold secondary antibodies, before being imaged by transmission electron microscopy. White arrows mark rhoptry bulb (RB) and neck (RN), while black arrows mark PfCERLI1 foci.”**

The following section has been added to the results in relation to the ImmunoEM results:

(line 267-270) **“To confirm these findings, we performed transmission electron microscopy (TEM) of immunogold labelled PfCERLI1^{HAGImS} compound 1 treated schizonts. Supporting immune-fluorescence localisation experiments, PfCERLI1 foci localised towards the periphery of the rhoptry bulb (Fig. 4g).”**

The following section has been included in the methods:

(line 739-810) **“Immunogold labelling and transmission electron microscopy**

Schizont stage parasites were Percoll purified and fixed in 1% glutaraldehyde in RPMI-HEPES buffer. Samples were dehydrated in increasing concentrations of ethanol and embedded in LRGold resin. Ultrathin sections were cut and then labelled with anti-HA antibody (Mouse monoclonal clone 12CA5, Roche, diluted 1:500) and then with 18 nm Colloidal Gold (AffiniPure Goat Anti-Mouse IgG, product 115-215-166, Jackson ImmunoResearch, diluted 1:20). Sections were then post-stained with uranyl acetate and lead citrate and viewed using a Talos L120C Transmission Electron Microscope.”

9. In Figure S6 where 3D super-resolution microscopy is used to measure merozoite proteins what does n=5 mean in these experiments? 5 merozoites? 5 schizonts, where all merozoites within a schizont were averaged? The error bars for some of the measurements deemed to be significantly different are exceedingly small, which makes me worry that these are only technical replicated and there may be batch effects.

Response: The data presented in Supplementary Figures 6 and 8 came from 5 independent biological replicates. Within each of those biological replicates, 10-15 images of compound 1 treated schizonts were captured. From each of these images the foci of either RAP1 (Supp Fig. 6b), RON4 (Supp Fig. 6d) or HA (Supp Fig. 8c)

were quantified, giving over 1000 measured foci (5 replicates X 10-15 schizonts X 20 merozoites per schizont) for each antigen in each condition. This high number of quantified foci is, therefore, the reason why the error bars appear so small and why seemingly small differences were sometimes highly statistically significant. We have clarified the number of foci measured by showing the individual datapoints on each graph in Supplementary Figures 6 and 8, rather than bar graphs, and modified the figure legends to reflect this as follows:

Supplementary Figure 6b

(supplementary line 249-250) "n=5 biological replicates, 1428 RAP1 foci counted for untreated parasites and 1243 for + 2.5 mM GLCN parasites"

Supplementary Figure 6d

(supplementary line 253-255) "n=5 biological replicates, 2962 RON4 foci counted for untreated parasites and 1939 for + 2.5 mM GLCN parasites"

Supplementary Figure 8c

(supplementary line (288-289) "(n = 5 biological replicates, 1489 PfCERLI1^{HA} foci counted for untreated parasites and 1197 counted for + 2.5 mM GLCN parasites)."

For further clarity regarding the colocalisation analyses (Figure 5, Supplementary Figure 7) and measurements of rhoptry diameter (Figures 4 and 5), we also included the number of analysed parasites/foci in the figure legends as follows:

Figure 4e

(line 1111-1112) "(n = 5 biological replicates, 1139 rhoptries measured for PfCERLI1^{HA} and 1040 for RAP1)."

Figure 5b,c & d

(line 1131-1132) "n=5 biological replicates, 38 schizonts counted for untreated and 36 for + 2.5 mM GLCN schizonts"

Figure 5e

(line 1134-1135) "n=5 biological replicates, 1220 rhoptries counted for untreated and 1217 for + 2.5 mM GLCN"

Supplementary Figure 7

(supplementary line 272-273) "Colocalisation was calculated for 19 untreated schizonts and 21 + 2.5 mM GLCN schizonts."

10. In this same experiment, how do you control for variation in developmental stage of a “schizont”? In the representative images in S6a, based on the number of merozoites that can be distinguished it looks like the GLN treated schizont is older than the untreated one. Perhaps these measurements need to all be done on schizonts with equal numbers of merozoites, and biological replicates should include merozoites from multiple schizonts.

Response: In our analyses, the parasites were synchronised to an approximately 4 hour window and treated with compound 1 to prevent rupture. In this way, we allowed tightly synchronised schizonts to mature to the same stage using the most practicable method available in the field. We then combined this synchronisation with the analysis of >1000 merozoites per treatment, over 5 biological replicates. Additionally, for each biological replicate, both treatments were derived from the same initial culture. We selected schizonts for imaging based on (1) clear nuclei segmentation (typically around 20 per schizont) and (2) clear RAP1/RON4 signal, which would indicate apical organelle maturation and effective antibody staining. Both assessment of schizont rupture and counts of the number of merozoites per schizont indicate that GLCN treatment did not have a significant effect on merozoite development. Given this, we believe that the imaging data was accurately assessed using a representative population in both GLCN treated and untreated parasite populations, and less developed schizonts will be equally represented in both. We agree that the representative image for Fig. 6c may suggest that the parasites in this treatment are less well developed, but this individual image is not representative of the population as a whole. Indeed, presenting this one image highlights the strength of our analysis over thousands of parasites, since comparison between a small number of cells can always be compromised by a small number of non-representative phenotypes. In order to avoid confusion, we have replaced this image with one that has similar numbers of merozoite nuclei for both the +/- GLCN treatments in Fig. 6a and the GLCN – GLCN treatment in Fig. 6b. It should be noted that schizonts of an identical stage of development (as indicated by nuclei number) were equally distributed across both +/- GLCN treatments. The replacement image clearly demonstrates knockdown of PfCERL1 protein expression, as expected, as well as showing clear staining of RON4 that was used in the quantitative analysis and selection criteria for schizont imaging, further decreasing the likelihood of developmental differences.

11. In the western blot in Figure 5F-G there is not a convincing difference between the amount of RON4 observed in pellet/supernatant in the untreated versus treated parasites. Also, I believe there is some evidence to suggest that RH5 is actually a micronemal protein (not rhoptry), so this is not an ideal protein to examine in this experiment. Given the very mild phenotype seen for RON4 it would be helpful to include some other rhoptry neck and/or bulb proteins in this experiment in order to provide more support for the conclusion that PfCERL1 knockdown impairs rhoptry content release.

Response: Given the suggestion that PfRH5 may localize to the micronemes, and the fact that the size of the protein resulted in the PfRH5 band being altered by the

high concentration of albumin in the secreted fraction, we have rerun western blots instead probing with a PfRH4 monoclonal antibody. PfRH4 is considerably larger and so is not influenced by the presence of albumin in the secreted fraction. With PfRH4 we see PfCERLI1 knockdown resulting in a 41% reduction in secreted PfRH4. Figures 5f and g have been updated to reflect the replacement of PfRH5 data with that of PfRH4.

12. If there is truly a decrease in RON4 release from the rhoptry, this would imply that the moving junction would not form normally, as the RON complex needs to be introduced into the uninfected RBC for formation of the moving junction. To more directly address this question I suggest looking directly at formation of the moving junction in invading merozoites that have or have not been treated with GLCN, or at localization of the components of the moving junction during invasion (RON complex and AMA1). This would provide more compelling evidence that problems with rhoptry antigen discharge are the reason for impaired invasion. If this is deemed out of the scope of this work at this time, then I suggest that the conclusion that PfCERLI1 plays a key role in controlling the distribution and secretion of rhoptry proteins during invasion should be dampened and presented as more of a working hypothesis in the discussion.

Response: As described in point 5, we attempted to use CytoD to quantitatively address whether PfCERLI1 knock-down blocked invasion prior to formation of the tight-junction, as would be expected when secretion of rhoptry neck proteins such as Rh4 (likely involved in receptor ligand interactions prior to tight-junction formation) and RON4 (part of the AMA1/RON complex involved in forming the tight-junction) is inhibited. However, this experiment proved non-informative as continuous treatment with CytoD did not lead to an increase in merozoites bound to the RBC membrane, as reported for short (2 minute) treatments of free merozoites (Riglar et al. Cell Host & Microbe, 2011). Hence, our control to assess tight-junction formation did not work in this setting. As outlined in point 5, quantitative comparison of bound merozoites and the tight/moving-junction using free merozoites is not feasible due to the inability of standardizing between parasite populations. However, we have been able to show that knock-down of PfCERLI1 also leads to aberrant processing of the rhoptry antigen PfRAP1, and present this data as a wider indication that PfCERLI1 has an important role in maintaining the correct function of the rhoptry, although the exact mechanisms need further investigation. Given we see an increase in the number of free merozoites which is proportional to the reduction in successful invasions (ring stages), but have not been able to show the block in invasion is at or before tight-junction formation *per se*, we agree with the reviewers suggestion and present in the discussion that PfCERLI1 plays a role in rhoptry function as a working hypothesis:

Concluding sentence

(line 495-503) "Taking all the data obtained together, we propose that PfCERLI1 has an important role in the correct function of the rhoptry during invasion, with knock-down of PfCERLI1 leading to abnormal distribution, secretion and processing of rhoptry antigens that are likely to be essential for merozoite invasion into the host"

erythrocyte. While further studies need to be undertaken to determine the fine detail of how PfCERLI1 knock-down causes these changes in rophtry function, identification of PfCERLI1's direct association with release of rophtry antigens is a key step in understanding the complex molecular events that control rophtry secretion during invasion."

Reviewer 1 minor comments:

1. **Fig 1c can be deleted and instead the reference for SLI-TDG paper can just be provided, especially since the technique was unsuccessful for PfCERLI1 KO in this case.**

Response: We have moved the schematic for the unsuccessful SLI-TGD knockout to Supplementary Figure 1a. Given that this technique was published relatively recently (April 2017), we believe some readers may not be familiar with it and therefore the schematic provides a simple understanding for how the system works and would like to keep it as part of the manuscript (in the supplementary figures).

2. **Figure panel 1d would be more informative if it showed a schematic of the integration plasmid and site of homology where recombination is expected to occur, along with binding location of primers used to confirm correct integration.**

Response: We have clarified this figure panel by making the integration sites more obvious. Additionally, we have included the binding locations of the primers used to confirm integration in Figure 1c and also in Supplementary Figure 1b where we confirm the integration of the Pfcerli1^{HAGImS} construct.

3. **Figure 1g: Does n=3 refer to technical replicates or biological replicates? If biological replicates, please define. Please indicate the ring parasitemia detected in the media control condition in the legend**

Response: n=3 in Figure 1g refers to biological replicates, where each biological replicate value is the mean of technical triplicates. The number of replicates, along with the inclusion of media control parasitaemia, has been clarified in the figure legend as follows:

(line 1046-1053) "(g) Synchronous PfCERLI1^{HAGImS} or 3D7 trophozoite-stage parasites were treated with increasing concentrations of GLCN for 48 hours, with the number of trophozoites the following cycle measured to determine knockdown-mediated growth inhibition (Each data point represents the mean of triplicate wells from three biological replicates. Parasite growth expressed as a % of non-inhibitory media controls with a mean parasitaemia of ~7%, error bars = standard error of the mean (SEM)). X-axis presented as a log 2 scale for viewing purposes."

4. **For Figure 2a, does n=4 indicate technical replicates or biological replicates? If biological replicates, please indicate how these are defined (different parasite clones, same clone run on different days, etc?).**

Please indicate the % ring parasitemia detected at 24 hours in the control condition in the legend.

Response: n =4 in Figure 2a refers to biological replicates, where each biological replicate value is the mean of technical triplicates. The number of replicates, along with the inclusion of media control parasitaemia, has been clarified in the figure legend as follows:

(line 1056-1064) **“Fig 2. PfCERLI1 knockdown does not inhibit merozoite development but does prevent merozoite invasion. (a) Flow cytometric detection of GFP-expressing PfCERLI1^{HAGlmS/GFP} ring stage parasites after merozoite invasion indicated a direct inhibition of merozoite invasion with protein knockdown (results presented as a % of media control, each data point represents the mean of triplicate wells from four biological replicates). Mean ring stage parasitaemia of media controls = ~5%. PfCERLI1^{HAGlmS} Growth is replicated from Figure 1g for direct comparison between growth and invasion inhibition. X-axis presented as log 2 scale for viewing purposes.”**

Reviewer 2 major comments:

- 1. The authors see an 80% reduction in protein levels by Western Blot by 48h. This is of course a bulk measure. How does the reduction look in individual cells? An IFA to look at depletion of PfCERLI1 levels in individual cells is important. Do different cells have different levels of protein knock-down? Or, do all cells show the same level of reduction? This result has important implications for the phenotype.**

Response: Quantitative IFA data looking at PfCERLI1 levels within individual merozoites is shown in Supplementary Figure 8c (Total PfCERLI1^{HA} intensity), however, it was presented as a bar graph and so the distribution of signals from individual merozoites could not be seen. To address this, we have changed this graph to show the signal intensities of each detected foci. Additionally, we have changed all the other bar graphs measuring foci in supplementary Figures 6 and 8 to showing all data points. >95% of glucosamine treated PfCERLI1^{HA} parasites show HA-staining at a signal intensity less than half the mean of untreated parasites. This demonstrates by far the majority of PfCERLI1^{HAGlmS} parasites show a major reduction in PfCERLI1 protein expression with GLCN treatment. In order to reflect this, we have added the following statement in the results when discussing PfCERLI1 protein knockdown:

(line 130-139) *“To validate that the integrated GlmS ribozyme could control PfCERLI1 protein expression, we treated PfCERLI1^{HAGlmS} parasites with 2.5 mM GLCN and quantified changes in HA-tagged protein levels using Western blot. Treatment of synchronous parasites with 2.5 mM GLCN for ~44 hours from early ring stage led to a >80% reduction in PfCERLI1 expression, whereas expression of the loading control EXP2 was not affected (Fig. 1f; Supplementary Fig. 1b,c).*

Additionally, immunofluorescence microscopy analysis revealed a significant reduction in HA labelling across nearly the whole PfCERLI1^{HAGImS} parasite population, with a reduction of the HA staining by ~97% (GLCN treated vs the mean of untreated parasites, Supplementary Figure 8c)."

- 2. The authors conclude from the increased number of free merozoites in culture supernatants that a defect in invasion prior to tight junction formation exists (lines 175-176). The authors also note that microneme secretion (as measured by EBA175 secretion) is not compromised in these parasites. Microneme secretion of certain proteins (including EBA175) establish secondary interactions with the RBC. It would be very useful to decipher exactly at what step, erythrocyte invasion is compromised in the PfCERLI1 knock-down parasites. Several questions arise: Do these merozoites attach to RBCs and then fall off? What degree of attachment do they have? Measurement of attachment of the PfCERLI1 KD merozoites to erythrocytes, by attachment assays AND/OR video microscopy to score for deformation of the RBCs by these merozoites would be valuable additions.**

Response: As outlined in the response to Review 1 (Comment 5). Quantitative assessment of bound merozoites using purified merozoites and fluorescence microscopy between two different parasite populations (+/- GLCN) is inherently difficult owing to the complete inability to standardize parasites populations and overall viability after filtering whole schizonts through a 1.2 µM filter. We did make a concerted attempt to assess this possibility using CytoD treatment of rupturing cultures to assess potential reductions in bound merozoite populations, but found continuous treatment of schizonts with CytoD did not lead to an increase in bound merozoites, possibly due to early prevention of acto-myosin motor engagement Reviewer 1 (Comment 5). Owing to the inability to get the CytoD control to work effectively, we have not included this data in the manuscript. In terms of live-cell microscopy, we achieved a ~40% knock-down in parasite invasion with GLCN treatment. After discussing using this approach with Dr Paul Gilson (Burnet Institute, Melbourne) who has led the application of live-cell imaging of merozoite invasion to assess invasion defects (Weiss et al, PLoS Pathogens 2015; Alanine et al. Cell, 2019; Wilson et al, BMC Biology, 2015), we reached the conclusion that the invasion defect using the glucosamine inducible riboswitch line (~40% reduction in invasion) was not likely to be high enough to lead to clearly quantifiable and interpretable analysis of invasion defects. Generally, Dr Gilson is looking for invasion inhibition of >80% before quantitative live filming assessment is practicable (*Personnel Communication*). Therefore, we have not undertaken these highly specialized and time-consuming experiments for this manuscript, but we are planning on trialing these assays in the future using a diCre mediated knock-down system that is likely to provide a stronger growth defect for PfCERLI1.

- 3. It has been previously observed that treating parasites with cytochalasin-D blocks entry but does not prevent rhoptry secretion -**

whorls of RAP1 can be seen in uninvaded RBCs (see Fig 6, Riglar et al (2011) Cell Host & Microbe). If an IFA were performed on the RBCs, which the PfCERLI1 knock down merozoites failed to invade, would one see rhoptry proteins aberrantly secreted into the RBCs? If this experiment gives a negative result compared to the positive control, it would strengthen the idea that PfCERLI1 knock downs do not form a tight junction.

Response: We agree this direct observation of the alterations to rhoptry secretion would provide extremely strong evidence that tight junction formation is inhibited, however as detailed for Reviewer 1 (Comment 5), it would be extremely difficult to get a quantifiable phenotype using current methods. Ideally, we could prepare two lots of purified merozoites either GLCN treated or untreated, add them to RBCs and after a few minutes add cytochalasin D and fix as in Riglar *et al.*, 2011 (Cell Host & Microbe). It would be nearly impossible, however, to simultaneously magnet purify out GLCN treated and untreated parasites and subsequently prepare free merozoites without uncontrollable differences in both merozoite number and viability. An alternative quantitative analysis would be to add cytochalasin D to late stage schizonts where the resulting merozoites would be inhibited at the point of junction formation and assess whether PfCERLI1 knock-down leads to a reduction in bound merozoites. Unfortunately, as outlined in response to Reviewer 1 (Major Comment 5), our attempts to use cytochalasin-D treatment to anchor merozoites to the outside of the RBC (bound merozoites) on the scale of hours that would be required for this kind of experiment was unsuccessful.

In response to these and the comments of Reviewer 1, we have revised our wording around this discussion as follows.

Results

(line 176-186) “When the number of lost invasion events (lost ring stage parasitaemia) was subtracted from the number of free merozoites, there was no difference between GLCN treated and untreated parasites ($p>0.99$, Supplementary Fig. 2c), indicating that the increase of the free merozoite population was proportional to the loss in successful merozoite invasion events. Additionally, GLCN treated schizonts appeared morphologically normal when Giemsa stained (Fig. 2e), suggesting PfCERLI1 knockdown does not result in schizont developmental defects. These data indicate that knockdown of PfCERLI1 is associated with a build-up in the number of free merozoites in culture, a pattern consistent with a possible early invasion defect prior to formation of the tight-junction and engagement of the actomyosin motor.”

Discussion

(line 461-469) “Knockout and knockdown studies implicate PfCERLI1 as having an important role in asexual stage parasite growth and we further confirmed that loss of this protein led to a defect in invasion. Initial investigation of the PfCERLI1 knockdown parasites showed that the number and morphology of merozoites was unaffected by loss of PfCERLI1 and parasites that did invade with PfCERLI1 knockdown developed normally. We determined that PfCERLI1 knockdown led directly to a significant reduction in newly invaded ring stage parasites and a

proportional build up in free merozoites in the culture, possibly through stopping invasion at a point prior to the formation of the tight junction and engagement of the acto-myosin motor.”

Reviewer 2 minor comments:

1. **Colocalisation analysis indicated that PfCERLI1 is most closely associated with the rhoptry bulb protein RAP1. The authors, through extensive biochemical analysis, then show that PfCERLI1 is peripherally associated with rhoptries. Together with this, immuno-EM studies would prove beyond doubt that PfCERLI1 is at the outer leaflet of rhoptries. The authors should consider doing this experiment.**

Response: As suggested by Reviewer 1 (Comment 8) and 2, we have undertaken the immuno-EM assessment of PfCERLI1 and this high resolution analysis supports that PfCERLI1 lies on the cytosolic face of the rhoptry bulb. To incorporate this new data, we have modified Figure 4 to include the Immuno-EM data:

(line 1115-1119) **“Fig 4. PfCERLI1 is peripherally-associated with the cytosolic face of the rhoptry membrane. (g) Representative image of Compound 1 treated PfCERLI1^{HAGImS} schizonts that were fixed, labelled with anti-HA antibodies and probed with 18 nm colloidal gold secondary antibodies, before being imaged by transmission electron microscopy. White arrows point to the rhoptry bulb (RB) and rhoptry neck (RN), while black arrows mark PfCERLI1 foci.”**

2. **Lines 140-141 “By contrast, GLCN treatment did not lead to significant growth inhibition in control 3D7 parasites.” Is this data available to be shown in the supplement? The authors say that 2.5 mM GLCN has “minimal non-specific growth inhibitor activity” – a quantitative statement for this would be beneficial.**

Response: The data this statement was based on has been included in Supplementary Figure 2a. Briefly, 3D7 parasites were treated with increasing GLCN concentrations, from ring stages, for one or two growth cycles at either 2.5 mM or 5 mM. In this assay, 2.5 mM GLCN had a negligible effect on parasite growth even over two growth cycles, while 5 mM GLCN reduced growth by ~ 35% over 2-cycles. The minimal reduction in growth for the longer 2-cycle treatment time at 2.5 mM further demonstrates that this concentration of GLCN has minimal non-specific growth inhibitory effects. We have modified the section in the results to say:

(line 145-148) **“At 2.5 mM GLCN, which we found to have minimal non-specific growth inhibitory activity against 3D7 WT parasites even after 96 hours of parasite treatment (Supplementary Fig 2a), PfCERLI1^{HAGImS} blood-stage parasite growth was reduced by 40% after 48 hours treatment.”**

To reflect the inclusion of this data, the figure legend has been modified as follows:

(supplementary line 194-199) **“Supplementary Figure 2. Influence of GLCN on wildtype parasite growth, PfCERLI1 knockdown on parasite development post-invasion, and the contribution of invasion inhibition to quantified free merozoites. (a) 3D7 wildtype parasites were treated with increasing concentrations of GLCN for 96 hours, to assess off-target growth inhibitory effects of GLCN (parasite growth expressed as a of media control (n = 5, error bars = SEM).”**

- 3. In Fig. 2d, the number of free merozoites are only marginally higher (1-2 %?) than the controls? Do these fully account for the 40% growth defect seen in the knock down?**

Response: Typically, only a relatively small proportion of released merozoites go on to invade new RBCs in culture. Here, based on the replication rate (5.5 fold per cycle) and the mean number of merozoites that develop per schizont (~19) we estimate that only ~30% of produced merozoites successfully invade. The remaining ~70% of merozoites that fail to invade can be measured free in the media by flow cytometry. We found that GLCN treatment of PfCERLI1^{HAGImS} parasites stops ~40% of invasion events at 2.5 mM, resulting in only ~18% of the merozoites produced actually invading into new RBCs. In Figure 2d, we show that there is an ~17% increase in the number of free merozoites with GLCN treatment (no GLCN 7% (normalised to number of RBCs), 2.5 mM GLCN 8.2% (normalised to number of RBCs); $9.2/7=1.17$). Thus, the increase in free merozoite number we would predict to occur with a 40% block in invasion prior to tight junction formation is reflected in the flow cytometry assessment of increased free merozoite numbers.

- 4. In supplementary Figures 1 and 2, the legends have been interchanged – please correct this oversight.**

Response: This has been corrected.

- 5. Line 156 – what is an early trophozoite? This is a long window of development – the number of hours should be mentioned for an accurate descriptor.**

Response: This has been clarified in the text by including the time, post-invasion, of treatment as follows:

(line 157-161) “Since PfCERLI1 was expressed only at schizont stages, we limited the amount of time PfCERLI1^{HAGlms/GFP} parasites were exposed to increasing concentrations of GLCN (0.125 to 5 mM) by only treating from early trophozoite stages (24 hpi) for 24 hours before assessing parasite rupture and invasion.”

- 6. Compound 1 – The authors make use of Compound-1 to arrest schizonts at a very mature stage without letting them undergo egress. Some descriptive sentences about what Compound-1 is, and some references for its use in prior literature are appropriate and necessary.**

Response: We have included a more detailed explanation of Compound 1 in lines 249-254, the first time it is mentioned, as follows:

(line 254-259) “In order to assess whether PfCERLI1 is also associated with markers of the rhoptry neck that lie closer to the apical tip of the merozoite, we triple antibody-labelled schizonts that had been treated with compound 1, a Protein kinase G inhibitor that prevents parasitophorous vacuole membrane rupture²⁹, with anti-HA (PfCERLI1), anti-RAP1 (rhoptry bulb), and anti-RON4 (rhoptry neck) antibodies prior to super-resolution microscopy.”

29 = Taylor, H. M. et al. The Malaria Parasite Cyclic GMP-Dependent Protein Kinase Plays a Central Role in Blood-Stage Schizogony. *Eukaryotic Cell* **9**, 37, doi:10.1128/EC.00186-09 (2010).

Reviewers' Comments:

Reviewer #1:

Remarks to the Author:

The authors have constructively and satisfactorily responded to the majority of the reviewer comments and suggestions, and should be commended for their attention to detail in the revised manuscript.

1. For figure 2D, I still have some reservations about the use of "free merozoite percentage" as a way to understand an invasion defect, as free merozoites in a population of red cells are not routinely measured by flow cytometry (due to their small size). To lessen this concern, it would be helpful for the authors to show representative flow cytometry plots for the gating strategy used for these experiments in a supplemental figure. In the same supplemental figure, I suggest they provide the math that they detail in response to reviewer 2, #3. This will help readers understand the interpretation that the free merozoite accumulation is "proportional" to the invasion defect.

2. I do not see a source data file with raw data for the invasion and growth assays. Some numerical data are provided, but they are only for one of the assays (Figure 1g?) and are not raw data but rather are normalized relative to control. To satisfy the journal's request for source data file with raw data, the authors should provide raw parasitemia data for figure 1g and 2a.

3. I agree with the authors that they need to cite and discuss the Suarez et al paper since it was published several months ago now and unfortunately dampens the novelty of the submitted manuscript. Including a paragraph in the discussion highlighting the overlaps and differences as they suggest is appropriate.

Response to Reviewer's Letter for Manuscript: 'PfCERLI1 is a conserved rhoptry associated protein essential for *Plasmodium falciparum* merozoite invasion of erythrocytes.' by Liffner B., et al.

Response to Reviewer's Comments

We thank the Reviewer and Editor for their feedback. We provide a description of the changes made to the manuscript to address the second round of feedback from Reviewer #1. All changes can be cross-referenced following the line numbers listed in either the main text or supplementary materials.

Reviewer #1 comments:

1. For figure 2D, I still have some reservations about the use of “free merozoite percentage” as a way to understand an invasion defect, as free merozoites in a population of red cells are not routinely measured by flow cytometry (due to their small size). To lessen this concern, it would be helpful for the authors to show representative flow cytometry plots for the gating strategy used for these experiments in a supplemental figure. In the same supplemental figure, I suggest they provide the math that they detail in response to reviewer 2, #3. This will help readers understand the interpretation that the free merozoite accumulation is “proportional” to the invasion defect.

Response: We have included representative flow cytometry plots, as Supplementary Figure 10, showing the gates used to for both newly invaded ring-stage parasites and free merozoites. Additionally, we have included representative plots of both GLCN untreated and 2.5 mM GLCN treated PfCERLI1^{HAGImS} parasites, where a decrease in ring-stages and an increase in free merozoites can be seen in treated parasites relative to untreated.

The figure legend of Supplementary Figure 10 is as follows (Supplementary information line 302-308):

“Supplementary Figure 10. Representative flow cytometry gating plots for invasion assays. PfCERLI1^{HAGImS/GFP} free merozoites were gated as Ethidium bromide^{high} FSC-H^{low} events in the ungated sample. PfCERLI1^{HAGImS/GFP} ring stage parasites were gated as Ethidium bromide^{low} GFP^{high} events inside the erythrocyte gate. GLCN-mediated knockdown of PfCERLI1 inhibited merozoite invasion, leading to an increase in the free merozoite population and a decrease in the ring stage population.”

Reference to this gating strategy has also been made in the methods section as follows (Line 821-836):

“To assess the impact of PfCERLI1 knockdown, 3D7 Δ Pfcerli1^{HAGImS} (growth) and 3D7 Δ Pfcerli1^{HAGImS/GFP} (invasion) parasites were synchronised to ring stages using sorbitol lysis and assays were set up in 96-well U-bottom plates at 1% parasitaemia and 1% haematocrit in a volume of 45 μ L⁵⁶. 5 μ L of 10x concentration GLCN or complete media was added to make a final volume of 50 μ L. Assays were stained with 10 μ g/mL ethidium bromide (Bio-Rad) in PBS before assessment of parasitaemia using flow-cytometry (BD Biosciences LSR II, 488 nm laser with FITC and PE filters). To assess merozoite development and invasion of 3D7

ΔPferli1^{HAGImS/GFP} parasites, GLCN treated and untreated cultures were grown for 36 hours, until newly invaded rings were present (0-6 hours post-invasion). Flow cytometry data was analysed using FlowJo (Tree Star). The gating strategy used to identify free merozoites and ring-stage parasites can be seen in Supplementary Figure 10.”

Additionally, the maths explained in response to reviewer 2 comment #3 is detailed as follows in the supplementary methods (Supplementary information line 19-32):

“Theoretical calculation of invasion inhibition contribution to free merozoites

To calculate the build-up of free merozoites that can be detected relative to control due to invasion inhibition, the number of free merozoites was determined using flow cytometry as described above and the following equations were implemented, with the final equation applied to each experimentally determined value:

$$\frac{\text{Parasitaemia fold change}^\#}{\text{Merozoites per schizont}^\circ} = \text{Prop. of produced merozoites that invade (mi)}$$

$$\frac{5.5}{19.1} = 0.288$$

$$\text{avg inv. inhib} \times \text{mi} = \text{Prop. of merozoites that invade with KD (msi)}$$

$$0.52 \times 0.288 = 0.155$$

$$\text{mi} - \text{msi} = \text{Prop. of merozoites that fail to invade due to knockdown (mfi)}$$

$$0.288 - 0.155 = 0.133$$

$$(1 - \text{mfi}) \times \text{actual free merozoites} = \text{theoretical merozoites if no inv. inhib.}$$

[#] as determined by ¹, [@] average value from Figure 2b used.”

2. I do not see a source data file with raw data for the invasion and growth assays. Some numerical data are provided, but they are only for one of the assays (Figure 1g?) and are not raw data but rather are normalized relative to control. To satisfy the journal’s request for source data file with raw data, the authors should provide raw parasitemia data for figure 1g and 2a.

Unfortunately, we were unable to locate the raw % parasitaemia for all replicates that were in the previous version of the manuscript. However, we were able to access the original flow cytometry data and reanalyse these experiments as well as include additional repeat experiments that had previously been performed. Both the normalised % control results and corresponding raw % parasitaemia results have now been provided in the updated Source Data file. To reflect the small changes seen with the replacement data, the graphs in Figure 1g and Figure 2a that use this updated data have been updated. Importantly, the trends shown by this data and thus conclusions drawn from this data have not changed. Included below is a side-by-side comparison of the old (left) and updated (right) graphs to demonstrate their similarity.

Old graphs

New graphs

3. I agree with the authors that they need to cite and discuss the Suarez et al paper since it was published several months ago now and unfortunately dampens the novelty of the submitted manuscript. Including a paragraph in the discussion highlighting the overlaps and differences as they suggest is appropriate.

We have included a version of the paragraph included in the cover letter, which has been slightly modified for conciseness. It is currently the second to last paragraph in the discussion and reads as follows (line 689-705):

“A recent study by Suarez et al.,¹² reported functional analysis of a homologue of PfCERLI1, which they termed Rhoptry Apical Surface Protein 2 (RASP2), in the related Apicomplexan parasite Toxoplasma gondii. They identified that TgRASP2 localises to the cytosolic face of the rhoptry, is essential for tachyzoite invasion of the host cell and confirmed that the C2 domain of this protein binds to the rhoptry membrane through interactions with phospholipids. In the studies of T. gondii,

TgRASP2 localised along the length of the tachyzoite rhoptries with foci at the rhoptry tip. Parallel studies in P. falciparum also identified an essential function for PfCERLI1 (called PfRASP2 in Suarez et al.,) in invasion using a rapamycin inducible DiCre knockdown system. Although these two studies are highly complementary in the functional analysis of this protein, our quantitative analysis of the localisation of PfCERLI1 differs to that described by Suarez et al., with PfCERLI1 maintaining a clear rhoptry bulb localisation using both fluorescence and immuno-EM localisation in our study and a rhoptry tip localisation reported by Suarez et al. It is possible that PfCERLI1 and TgRASP2 have different localisations/functions, as while these organisms are related, they display markedly different host and cellular tropisms. However, further investigation will be required to tease out the specifics of these differences.”